# M$^2$RC-Eval: Massively Multilingual Repository-level Code Completion Evaluation

## Abstract

Repository-level code completion has drawn great attention in software engineering, and several benchmark datasets have been introduced. However, existing repository-level code completion benchmarks usually focus on a limited number of languages ($<5$), which cannot evaluate the general code intelligence abilities across different languages for existing code Large Language Models (LLMs). Besides, the existing benchmarks usually report overall average scores of different languages, where the fine-grained abilities in different completion scenarios are ignored. Therefore, to facilitate the research of code LLMs in multilingual scenarios, we propose a massively multilingual repository-level code completion benchmark covering 18 programming languages (called **M$^2$RC-Eval**), and two types of fine-grained annotations (i.e., **bucket-level** and **semantic-level**) on different completion scenarios are provided, where we obtain these annotations based on the parsed abstract syntax tree. Moreover, we also curate a massively multilingual instruction corpora **M$^2$RC-Instruct** dataset to improve the repository-level code completion abilities of existing code LLMs. Comprehensive experimental results demonstrate the effectiveness of our M$^2$RC-Eval and M$^2$RC-Instruct.

## 1 Introduction

The emergence of Large Language Models (LLMs) specifically designed for code-related tasks has marked a significant advancement in code generation. The code LLMs (Roziere et al., 2023; Zheng et al., 2023; Guo et al., 2024a; Hui et al., 2024b) pre-trained on extensive datasets comprising billions of code-related tokens further revolutionize the automation of software development tasks, providing contextually relevant code suggestions and facilitating the translation from natural language to code. The generation capability of code LLMs opens up diverse applications in software development, promising to enhance productivity and streamline coding processes. As the field continues to evolve, it presents exciting opportunities for future developments and innovations in automated programming and code assistance.

The code completion task is crucial in modern software development, enhancing coding efficiency and accuracy by predicting and suggesting code segments based on context. Recent advancements in code LLMs (Bavarian et al., 2022b) have introduced sophisticated completion techniques, such as prefix-suffix-middle (PSM) and suffix-prefix-middle (SPM) paradigms, which can complete middle code segments given the surrounding context. However, the current benchmark (Ding et al., 2024; Liu et al., 2023a) mainly focuses on several programming languages. For example, the Cross-CodeEval (Ding et al., 2024) includes four languages (i.e., Python, Java, TypeScript, C#). Besides, existing benchmarks can only provide the average score among all samples, which cannot provide a language-specific evaluation for different programming languages based on their intrinsic structure. *Inspired by the multilingual in-file code generation benchmark MultiPL-E Cassano et al. (2022) and McEval (Chai et al., 2024), we create a massively multilingual repository-level code completion Evaluation benchmark called* **M$^2$RC-Eval** *to facilitate the research of the community.*

In this paper, as shown in Fig. 1, our M$^2$RC-Eval includes 18 programming languages with two types of fine-grained annotations (i.e., **bucket-level** and **semantic-level**), where each language contains 100 validation and 500 test samples, respectively. Specifically, for the bucket-level annotations,

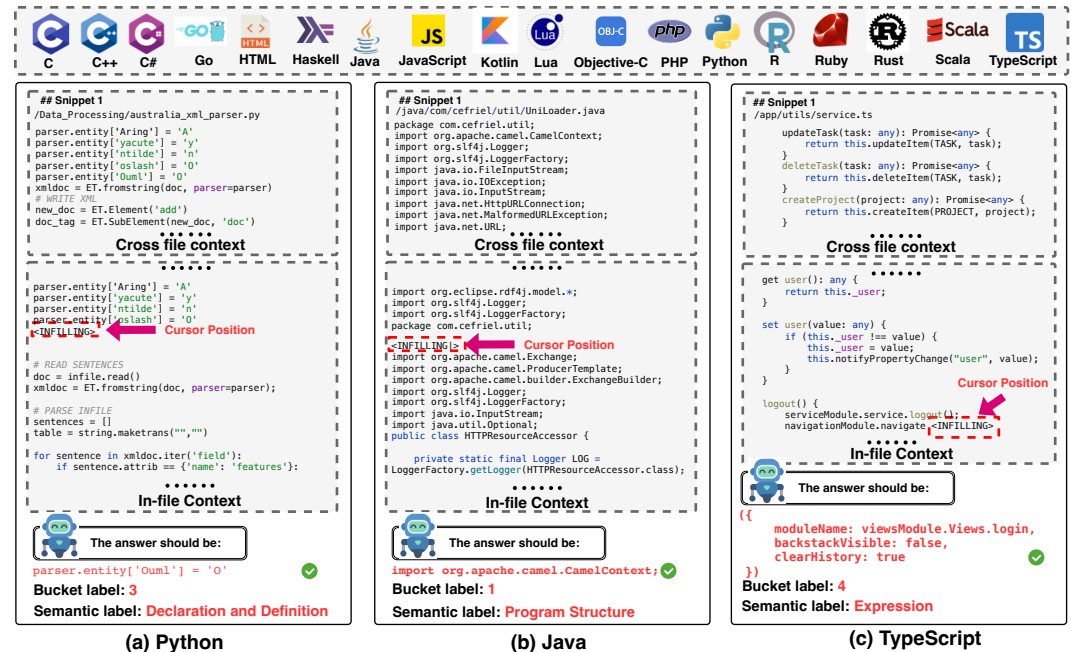

Figure 1: Overview of our proposed $M^2$RC-EVAL with 18 languages. Specifically, first, we provide three samples from different languages (i.e., Python, Java, TypeScript) for illustration, where the bucket label and semantic label for the corresponding cursor position are provided. Second, the code LLMs need to predict the completion results given the in-file context from the current code file and the cross file context retrieved from other code files in the current repository. Note that "< INFILLING >" denotes that the current position will be triggered for code completion.

we first generate abstract syntax tree with $N$ layers using code parser (i.e., Treesitter [1]), and divide these $N$ into fixed $M$ buckets, Then, for each completion cursor position, we annotate the corresponding bucket-level label based on the layer to which the location belongs. In this way, we can obtain different code completion scenarios with different difficulties.

For the semantic-level annotations, inspired by (Takerngsaksiri et al., 2024), we first pre-define 11 major semantic labels (e.g., Program Structure, Statement) for each completion cursor position, which aims to analyze the fine-grained performance across different code semantics. Note that as different languages usually have specific syntax, we carefully design the subcategories under each major semantic label for different languages. Then, as the code parser usually provides syntax labels (e.g., functions, variables, classes, empty lines)[2] for each completion cursor position, we carefully define the mappings between the syntax labels to our designed semantic labels and build the semantic-level annotations for our $M^2$RC-EVAL. Finally, to enhance the performance of repository-level code completion for existing code LLMs, we also create a massively multilingual instruction corpora **$M^2$RC-INSTRUCT** of 18 languages.

The contributions are summarized as follows:

- We propose the first massively multilingual repository-level code completion benchmark $M^2$RC-EVAL covering 18 languages, where two types of annotations (bucket-level and semantic-level labels) are provided based on the parsed abstract syntax tree.

- We introduce $M^2$RC-INSTRUCT, the massively multilingual repository-level code instruction corpora covering the multilingual code snippet from 18 languages, which can greatly enhance the performance of repository-level code completion results.

- Comprehensive evaluation results and analysis demonstrate the effectiveness of our proposed $M^2$RC-EVAL and $M^2$RC-INSTRUCT.

---

[1] https://tree-sitter.github.io/tree-sitter/

[2] Note that the syntax label provided by code parser (e.g., tree-sitter) are highly detailed.

## 2 RELATED WORKS

**Code Large Language Models.** Code large language models (LLMs) (Chen et al., 2021; Zhao et al., 2024; Black et al., 2021; 2022; Le et al., 2022; Chowdhery et al., 2023; Nijkamp et al., 2023; Fried et al., 2023; Xu et al., 2022; Jain et al., 2024b) are increasingly involved in modern programming, due to excellent capabilities of code generation (Li et al., 2022; Allal et al., 2023), code repair (Wang et al., 2021; 2023), code translation (Zheng et al., 2023; Li et al., 2023), and other coding tasks. UniCoder (Sun et al., 2024) and SPT-Code (Niu et al., 2022) introduce the pseudo-code generation and the alignment between Abstract Syntax Tree (AST) and code. Recent code LLMs such as Code Llama (Roziere et al., 2023), DeepSeek-Coder (Guo et al., 2024a), and Qwen2.5-Coder (Hui et al., 2024b) incorporate the fill-in-the-middle (FIM) task into their training stage for code completion. Moreover, there is a wide variety of in-file benchmarks to evaluate different capabilities of code LLMs (Zheng et al., 2023; Austin et al., 2021; Jain et al., 2024a), which focus on a limited range of programming languages (e.g. Python and Java). The recent work (Chai et al., 2024) extends the number of programming languages to 40 for multilingual evaluation scenarios, which has not considered the repository-level code completion.

**Repository-level Code Completion.** The latest repository-level code completion methods (Bairi et al., 2023; Phan et al., 2024; Liao et al., 2023; Shrivastava et al., 2023a; Agrawal et al., 2023; Shrivastava et al., 2023b; Pei et al., 2023; Zhang et al., 2023) are similar to RAG, aim to precisely retrieve all related code snippets across files within a repository. Further, repository-level benchmarks are proposed to estimate the capability of code LLMs in a more realistic software engineering scenario. But these datasets (Ding et al., 2023; 2022; Allal et al., 2023) are primarily concentrated on several programming languages. Regarding difficulty categorization, most methods only consider the number of files involved in the completion content, overlooking the code's structural and semantic context within the entire project. Repofusion (Shrivastava et al., 2023a) and Repocoder (Zhang et al., 2023) predict one line based on the prefix and suffix code, while CoderEval (Yu et al., 2024) measures how many third-party libraries are called. To comprehensively evaluate the multilingual repository-based code completion of different code LLMs, we push the boundaries of programming languages into 18 languages in $M^2$RC-EVAL with fine-grained annotations.

## 3 $M^2$RC-EVAL

### 3.1 DATA COLLECTION

**The Overall Data Pool.** We begin by collecting The Stack v2 (Lozhkov et al., 2024), which consists of permissively licensed repositories from GitHub. Next, we adopt the `The-stack-v2-dedup`, which includes 784 million source code files spanning 619 programming languages with manual and heuristic pre-processing. Further, we keep only repositories receiving more than 5 stars and containing $[10, 50]$ files. Lastly, preserving files written in 18 common languages, we have 431,353,244 files remaining, constituting the overall data pool.

**Completion Cursor Position Selection.** Completion cursor position selection significantly impacts the quality of a code completion benchmark. Previous studies (Ding et al., 2024; Liu et al., 2023a) randomly select a segment of consecutive characters as the completion span, which does not guarantee the integrity of identifiers and statements. Besides, recent works (e.g., Qwen2.5-Coder (Hui et al., 2024b), aiXcoder (Jiang et al., 2024)) also claimed that developers often expect LLMs to complete the current code into a complete snippet, such as a completed code line or loop block, instead of suggesting an incomplete code snippet. Therefore, in $M^2$RC-EVAL, we first parse the abstract syntax tree (AST) of each source code file, and then we randomly choose a node (e.g., the node of "Function Definition" in Fig. 2) on the AST as the completion cursor position. After that, we obtain the corresponding code to obtain the ground-truth for the current completion cursor position. Finally, at inference, the code LLMs need to predict the current code span given the in-file and cross file contexts. Similarly, in training, we just use the ground-truth to supervise the tuning process of the code LLMs.

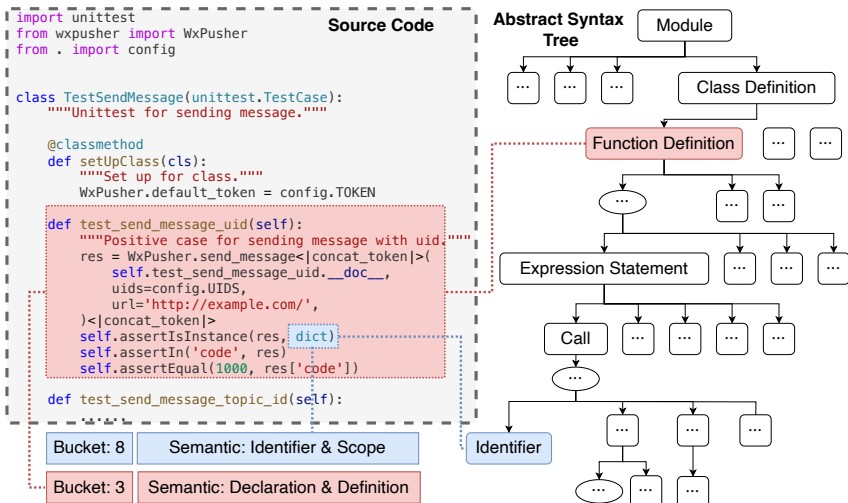

Figure 2: Illustration on generating completion cursor position and fine-grained annotations. Specifically, we first parse the source code into an abstract syntax tree (AST). Then, we choose one node as the completion cursor position and generate the bucket label based on the belonged layer number in AST, and obtain the semantic label based on the node type parsed by the Tree-sitter.

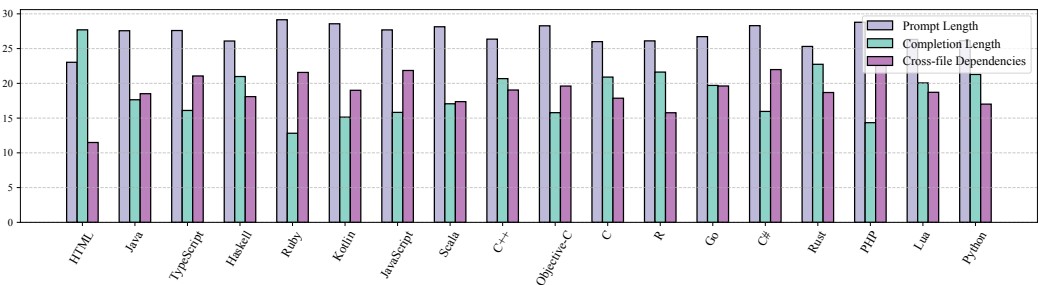

Figure 3: The average prompt length (100x tokens), completion span length (50x tokens), and cross-file dependencies (1x) in the testing set of $M^2$RC-EVAL. We define the number of other files, which are explicitly imported and implicitly referenced by the current file, as cross-file dependencies.

Table 1: A comparison with existing notable repository-level code completion datasets.

| Benchmark | # Languages | Fine-grained | Training Set | # Test Repos |
|---|---|---|---|---|
| RepoBench (Liu et al., 2023a) | 2 | ✗ | ✓ | 1669 |
| CrossCodeEval (Ding et al., 2024) | 4 | ✗ | ✗ | 1002 |
| $R^2C^2$-Bench (Deng et al., 2024) | 4 | ✗ | ✓ | 1353 |
| $M^2$RC-EVAL & $M^2$RC-INSTRUCT | 18 | ✓ | ✓ | 5993 |

## 3.2 QUALITY CONTROL

We build a suite of post-processing filters to enhance the quality of $M^2$RC-INSTRUCT. We eliminate examples based on two heuristic rules: (1) The completion cursor position should be no longer than 5 lines. (2) If the completion ground truth is fewer than 20 characters, at least 20% of them should be alphabetic. To improve data independence and inference difficulty, we apply extra filters to the test cases in $M^2$RC-EVAL. (a) Repositories in $M^2$RC-EVAL should be absent from $M^2$RC-INSTRUCT. (b) We ensure that 30% of the completion ground truth is not shorter than 2 lines. (c) The completion cursor position should not be fully white-spaced. (d) We discard test cases that could be exactly predicted by `DeepSeekCoder-1.3B` (Guo et al., 2024b) without cross file contexts.

Table 2: Semantic-level annotations on different types of programming languages.

| Major Classes | Java | Go | Scala |
|---|---|---|---|
| Program Structure | "Program Entry", "Namespace", "Import/Include" | "Program Entry", "Namespace", "Import/Include" | "Program Entry", "Namespace", "Import/Include" |
| Declaration and Definition | "Class", "Function", "Variable" | "Class", "Function", "Variable" | "Class", "Function", "Variable" |
| Control Flow Structure | "Conditional", "Loop", "Jump", "Exception Handling" | "Conditional", "Loop", "Jump", "Exception Handling" | "Conditional", "Loop", "Jump", "Exception Handling" |
| Expression | "Arithmetic Operation", "Logical Operation", "Function Call", "Object Creation", "Type Casting", "Other", "Arithmetic Operator", "Logical Operator" | "Arithmetic Operation", "Logical Operation", "Function Call", "Object Creation", "Type Casting", "Arithmetic Operator", "Logical Operator" | "Arithmetic Operation", "Function Call", "Object Creation", "Type Casting", "Tuple Expression", "Logical Operator", "Special Operator" |
| Data Type | "Primitive Type", "Composite Type", "Generic", "Numeric", "String", "Boolean", "Special Value" | "Primitive Type", "Composite Type", "Generic" | "Primitive Type", "Composite Type", "Generic", "Numeric", "String", "Boolean", "Special Value" |
| Statement | "Expression Statement", "Compound Statement", "Other Statement" | "Expression Statement", "Compound Statement" | "Compound Statement" |
| Modifier and Attribute | "Access Modifiers", "Other Modifiers", "Attribute Annotation" | "Access Modifiers", "Other Modifiers", "Attribute Annotation" | "Access Modifiers", "Other Modifiers", "Annotation" |
| Comments and Documentation | "Single-line Comment", "Multi-line Comment" | "Single-line Comment" | "Single-line Comment", "Multi-line Comment" |
| Preprocessing Directive | "Conditional Compilation", "Macro Definition" | "Conditional Compilation", "Macro Definition" | "Macro Definition" |
| Identifier and Scope | "Identifier", "Qualified Name" | "Identifier", "Qualified Name" | "Identifier", "Qualified Name", "Binding", "Delimiter" |
| Special Language Structure | "Lambda Expression", "Pattern Matching", "Coroutine" | "Lambda Expression", "Coroutine" | "Lambda Expression", "Pattern Matching" |

## 3.3 DATASET STATISTICS

Following the quality filters in §(3.2) from the overall data pool §(3.1). We sample 50,000 files per language to construct our M$^2$RC-INSTRUCT, and sample 100, and 500 files per language to build the validation and test sets of our M$^2$RC-EVAL, respectively. The statistics of the test set are shown in Fig. 3, and we also provide a detailed comparison between our M$^2$RC-EVAL with existing repository-level code completion datasets in Table 1. Note that the numbers of repositories for M$^2$RC-INSTRUCT, validation split of M$^2$RC-EVAL are 37439 and 1635, respectively.

## 3.4 FINE-GRAINED ANNOTATIONS

As shown in Fig. 2, to analyze the performance in a fine-grained manner, we further provide two types of fine-grained annotations (i.e., bucket-level and semantic-level) for each completion cursor. Specifically, we first generate the abstract syntax tree. For the bucket-level annotations, we first simply divide each tree into $M$ buckets based on the depth degree of the abstract syntax tree. Note that we set $M$ as 10 in our M$^2$RC-EVAL. For example, if the number of layers for the current abstract syntax tree is $N$, the $i$-th layer of the tree belongs to the $\lceil \frac{i}{N/M} \rceil$ bucket. Then, for each completion cursor node, we annotate the bucket label based on the layer number of each node. Similarly, for the semantic-level annotations, we directly annotate the semantic-level label for each completion cursor node. Specifically, we pre-define 11 major classes (i.e., "Program Structure", "Declaration and Definition", "Control Flow Structure", "Expression", "Data Type", "Statement", "Modifier and Attribute", "Comments and Documentation", "Preprocessing Directive", "Identifier and Scope", "Special Language Structure"). Then, as different languages have many specific designs, the subcategories under each major class are carefully annotated for different languages. As shown in Table 2, we provide the semantic-level annotations on three main-stream programming

Table 3: Exact match (%) and edit similarity (%) performance on M$^2$RC-EVAL.

| Model | C | | C# | | C++ | | Go | | HTML | | Haskell | | - | |
|---|---|---|---|---|---|---|---|---|---|---|---|---|---|---|
| | EM | ES | EM | ES | EM | ES | EM | ES | EM | ES | EM | ES | EM | ES |
| Code Llama-7B | 18.6 | 47.2 | 19.6 | 52.6 | 21.8 | 51.1 | 26.0 | 53.6 | 20.6 | 40.4 | 22.6 | 48.9 | - | - |
| + Retrieval | 21.8 | 47.2 | 22.9 | 48.9 | 23.2 | 46.6 | 23.8 | 52.4 | 12.6 | 35.6 | 22.6 | 48.9 | - | - |
| + Retrieval & Tuning | 45.4 | 72.0 | 43.5 | 72.3 | 50.8 | 74.9 | 43.4 | 72.9 | 41.8 | 63.6 | 39.8 | 66.3 | - | - |
| StarCoder-7B | 20.0 | 50.4 | 20.0 | 53.3 | 22.4 | 51.8 | 25.4 | 58.2 | 17.4 | 40.7 | 25.0 | 51.1 | - | - |
| + Retrieval | 23.8 | 47.8 | 27.1 | 53.2 | 24.6 | 48.0 | 26.0 | 53.6 | 20.6 | 40.4 | 25.0 | 47.7 | - | - |
| + Retrieval & Tuning | 47.0 | 72.7 | 45.1 | 74.8 | 52.4 | 76.3 | 43.2 | 73.7 | 45.8 | 67.1 | 44.8 | 70.2 | - | - |
| DeepSeekCoder-6.7B | 22.4 | 53.7 | 21.4 | 56.2 | 23.2 | 54.2 | 29.4 | 61.4 | 17.6 | 43.4 | 25.2 | 51.3 | - | - |
| + Retrieval | 28.2 | 52.6 | 25.3 | 52.6 | 27.6 | 52.2 | 29.4 | 61.4 | 17.6 | 43.4 | 25.8 | 51.0 | - | - |
| + Retrieval & Tuning | 48.6 | 75.2 | 47.9 | 76.9 | 54.4 | 78.2 | 48.8 | 78.4 | 45.0 | 66.3 | 45.8 | 72.0 | - | - |

| Model | Java | | JavaScript | | Kotlin | | Lua | | Objective-C | | PHP | | - | |
|---|---|---|---|---|---|---|---|---|---|---|---|---|---|---|
| Code Llama-7B | 23.4 | 58.5 | 17.2 | 52.0 | 23.6 | 57.0 | 20.0 | 45.7 | 17.8 | 49.5 | 19.2 | 54.9 | - | - |
| + Retrieval | 23.4 | 57.5 | 19.6 | 48.0 | 20.8 | 50.0 | 19.6 | 42.2 | 21.4 | 46.6 | 21.2 | 49.0 | - | - |
| + Retrieval & Tuning | 41.8 | 74.1 | 38.8 | 70.1 | 45.0 | 75.6 | 43.8 | 70.5 | 49.8 | 75.9 | 45.6 | 76.7 | - | - |
| StarCoder-7B | 24.0 | 59.2 | 16.6 | 52.0 | 24.4 | 59.3 | 21.4 | 48.6 | 17.6 | 49.6 | 18.6 | 54.4 | - | - |
| + Retrieval | 25.0 | 53.1 | 22.0 | 50.8 | 22.8 | 52.6 | 26.4 | 48.5 | 23.6 | 48.0 | 18.6 | 54.4 | - | - |
| + Retrieval & Tuning | 47.4 | 76.9 | 38.8 | 70.1 | 45.0 | 75.6 | 43.8 | 70.5 | 50.8 | 75.9 | 45.6 | 76.7 | - | - |
| DeepSeekCoder-6.7B | 22.2 | 61.0 | 20.4 | 56.5 | 26.0 | 61.0 | 22.0 | 48.8 | 21.0 | 55.6 | 24.2 | 58.6 | - | - |
| + Retrieval | 21.6 | 51.4 | 24.4 | 53.6 | 26.0 | 61.0 | 22.0 | 49.9 | 27.6 | 53.5 | 28.6 | 56.9 | - | - |
| + Retrieval & Tuning | 48.2 | 79.1 | 43.6 | 73.5 | 46.0 | 75.7 | 44.6 | 70.6 | 52.2 | 77.6 | 49.8 | 78.8 | - | - |

| Model | Python | | R | | Ruby | | Rust | | Scala | | TypeScript | | Avg. | |
|---|---|---|---|---|---|---|---|---|---|---|---|---|---|---|
| Code Llama-7B | 24.6 | 54.2 | 15.2 | 41.2 | 17.2 | 45.8 | 26.2 | 56.0 | 22.8 | 48.5 | 23.4 | 52.3 | 19.4 | 50.3 |
| + Retrieval | 17.4 | 46.4 | 15.2 | 39.8 | 17.2 | 42.3 | 26.0 | 51.3 | 22.8 | 48.5 | 19.4 | 48.6 | 20.2 | 46.1 |
| + Retrieval & Tuning | 39.2 | 69.9 | 38.6 | 65.5 | 43.0 | 68.5 | 42.0 | 69.2 | 41.0 | 70.1 | 37.0 | 68.2 | 41.9 | 70.0 |
| StarCoder-7B | 19.4 | 52.9 | 16.4 | 43.7 | 19.4 | 47.4 | 26.2 | 56.0 | 23.6 | 53.4 | 19.8 | 53.3 | 21.0 | 52.0 |
| + Retrieval | 24.6 | 54.2 | 22.6 | 47.2 | 23.6 | 47.4 | 26.4 | 53.5 | 22.8 | 48.5 | 23.4 | 52.3 | 24.1 | 50.0 |
| + Retrieval & Tuning | 39.2 | 69.9 | 41.0 | 66.6 | 43.0 | 68.5 | 45.8 | 72.6 | 43.6 | 71.5 | 39.2 | 69.7 | 44.5 | 72.2 |
| DeepSeekCoder-6.7B | 21.8 | 55.1 | 19.4 | 48.5 | 23.6 | 52.2 | 23.8 | 54.3 | 24.6 | 56.7 | 19.4 | 55.4 | 22.6 | 54.7 |
| + Retrieval | 21.8 | 55.1 | 19.4 | 48.5 | 23.6 | 52.2 | 23.8 | 54.3 | 22.4 | 50.4 | 26.0 | 54.5 | 25.1 | 51.7 |
| + Retrieval & Tuning | 41.6 | 71.3 | 45.4 | 69.4 | 45.6 | 70.3 | 47.6 | 73.4 | 44.8 | 73.7 | 43.2 | 73.4 | **46.8** | **74.1** |

languages (Java, Go, Scala), where the annotations on all 18 languages are provided in Fig. 12, Fig. 13 and Fig. 14 of the Appendix.

## 4 EXPERIMENTS

### 4.1 EVALUATION MODELS AND METRICS

We perform main experiments on M$^2$RC-EVAL with three Code LLMs (i.e., **StarCoder-7B** (Li et al., 2023), **DeepSeekCoder-6.7B** (Guo et al., 2024b) and **Code Llama-7B** (Roziere et al., 2023)) (See Appendix A.3 for more details). Following (Ding et al., 2023), we compare the generated code with the reference and compute the exact match (**EM**) and edit similarity (**ES**) metrics [3], which assess the textual similarities and ignore semantic structure similarities among predictions and ground-truth.

### 4.2 EXPERIMENTAL SETUP

**Baseline.** Only the original code file, where the cursor position is located, is provided for the code LLMs. As no explicit inter-file context is supplied, the model must utilize its inherent knowledge-based reasoning abilities to generate appropriate code.

**+ Retrieval.** In line with the approach outlined in CrossCodeEval (Ding et al., 2023), the retrieval process begins by examining files within the same repository. Continuous code segments of $L$ lines are extracted, where $L$ matches the length of the retrieval query and is set as 10 by default. Subsequently, these extracted candidates are prioritized based on their Jaccard similarity scores. The most relevant fragments are then appended to the beginning of the in-file context in descending

---

[3] https://github.com/amazon-science/cceval

Table 4: Performance on different LLMs on M$^2$RC-EVAL.

| Model | M$^2$RC-EVAL | | M$^2$RC-EVAL-2403 | | M$^2$RC-EVAL-2406 | |
|---|---|---|---|---|---|---|
| | EM | ES | EM | ES | EM | ES |
| Code Llama-7B | 19.4 | 50.3 | 19.1 | 52.9 | 21.5 | 52.7 |
| + Retrieval | 20.2 | 46.1 | 23.1 | 50.8 | 25.0 | 51.5 |
| StarCoder-7B | 21.0 | 52.0 | 20.4 | 53.1 | 20.1 | 51.6 |
| + Retrieval | 24.1 | 50.0 | 26.0 | 54.9 | 28.6 | 55.9 |
| DeepSeekCoder-6.7B | 22.6 | 54.7 | 20.4 | 51.9 | 23.0 | 55.6 |
| + Retrieval | 25.1 | 51.7 | 24.0 | 52.7 | 30.3 | 56.4 |
| DeepSeekCoder-33B | 26.8 | 51.6 | 24.0 | 43.7 | 23.9 | 49.7 |
| + Retrieval | 27.3 | 52.9 | 27.1 | 51.8 | 27.5 | 49.8 |
| Qwen2.5-Coder-7B | 18.8 | 46.5 | 20.5 | 49.7 | 21.0 | 48.1 |
| + Retrieval | 27.2 | 52.2 | 31.0 | 57.2 | 32.4 | 56.7 |
| Qwen2.5-Coder-32B | 34.7 | 65.7 | 35.0 | 66.2 | 37.3 | 67.6 |
| + Retrieval | 41.7 | 68.0 | 43.9 | 69.5 | 45.9 | 71.2 |
| LLama3.1-70B | 6.4 | 31.9 | 5.0 | 31.5 | 5.4 | 31.3 |
| + Retrieval | 6.8 | 33.0 | 6.1 | 33.3 | 6.1 | 32.8 |
| Qwen2.5-72B | 6.7 | 39.1 | 11.6 | 49.6 | 10.2 | 45.3 |
| + Retrieval | 12.2 | 44.8 | 12.4 | 51.1 | 13.4 | 50.9 |
| GPT-4o | 12.2 | 45.5 | 11.5 | 54.0 | 11.1 | 47.2 |
| + Retrieval | 17.8 | 56.7 | 15.0 | 57.4 | 17.3 | 54.0 |
| Claude 3.5 Sonnet | 22.4 | 55.3 | 23.2 | 63.8 | 23.1 | 59.5 |
| + Retrieval | 29.9 | 62.8 | 28.4 | 65.9 | 30.5 | 67.1 |
| DeepSeekV2.5 | 16.1 | 50.5 | 23.9 | 61.0 | 25.2 | 56.9 |
| + Retrieval | 27.2 | 60.6 | 28.3 | 64.1 | 26.0 | 61.1 |

order of similarity. This concatenation continues until the total length, including both the added candidates and the original in-file context, reaches the predetermined maximum token limit of 4096.

**+ Retrieval & Tuning.** To further improve the performance of repository-level code completion, we fine-tune code LLMs on the M$^2$RC-INSTRUCT dataset mentioned in §(3). At inference, we use the same inference strategy as discussed in "+ Retrieval".

## 4.3 MAIN RESULTS

We present the results on M$^2$RC-EVAL in Table 3. We observe that different code LLMs have different repository-level code completion abilities for different programming languages. For instance, DeepSeekCoder-6.7B demonstrates strong completion ability for Go, while its performance is weaker with HTML, a markup language, which demonstrates the necessity of evaluating code LLMs for multilingual capabilities. Besides, the results indicate that cross-file context is highly effective, resulting in a significant improvement compared to using only in-file context. In particular, the multilingual SFT on our created instruction corpora M$^2$RC-INSTRUCT also significantly enhances performance on M$^2$RC-EVAL. Notably, after SFT on M$^2$RC-INSTRUCT, Code Llama-7B, which originally ranked lowest with in-file context, outperformed the non-finetuned StarCoder-7B, demonstrating the effectiveness of M$^2$RC-INSTRUCT.

## 4.4 ANALYSIS

**Analysis on data leakage.** Following LiveCodeBench (Jain et al., 2024a) and EvoCodeBench (Li et al., 2024), we also build a dynamically updating M2rc-Eval dataset, where the **M$^2$RC-EVAL-2403** and **M$^2$RC-EVAL-2406** are produced in Table 4. Specifically, we collect repositories from 2024.03.01-2024.05.31 and then build the **M$^2$RC-EVAL-2403** split based on the same data collection process. Similarly, we build the **M$^2$RC-EVAL-2406** using repositories from 2024.06.01-2024.08.30, and the results (EM/ES) of different splits for different LLMs (Code Llama-7B, StarCoder-7B, DeepSeekCoder-6.7B, GPT-4o[4], LLama3.1 (Team, 2024a), Qwen2.5 (Team, 2024b),

---

[4]https://openai.com/index/hello-gpt-4o/

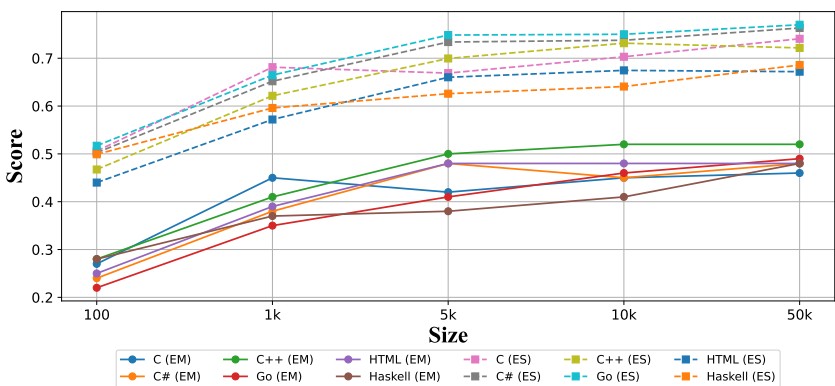

Figure 4: Effectiveness of using different training data sizes.

Claude 3.5 [5] and DeepSeek-V2.5 (DeepSeek-AI, 2024)) are provided. Note that as some models do not support the FIM pattern, we directly use a prompt engineering strategy to obtain the repository-level code completion results. We have the following observations. (1). When introducing cross-file context using retrieval, better performance results are usually obtained, specifically on the EM metric. (2). For existing Code LLMs, the performance on different testing splits is relatively stable, which means that data leakage or contamination concerns are almost non-existent in M$^2$RC-EVAL. Besides, for many knowledge-based benchmarks (e.g., MMLU (Hendrycks et al., 2020), SimpleQA (Wei et al., 2024)), this knowledge information widely exists in web and book corpus, which have been trained in existing LLMs. However, these benchmarks are still effective tools for evaluating the knowledge coverage degree in these LLMs. (3) Meanwhile, although our M$^2$RC-EVAL has been trained in several LLMs, we still find existing LLMs cannot achieve competitive performance results, and our M$^2$RC-EVAL can still be used as an effective benchmark to evaluate the code completion abilities of existing LLMs. (4) These powerful API LLMs or open-source LLMs (e.g., LLama3.1-70B, Qwen2.5-72B) have strong code generation abilities in many benchmarks (e.g., HumanEval (Chen et al., 2021), MBPP Austin et al. (2021), (Jain et al., 2024a)), the repository-level code completion performance are still limited when compared to these code-specific LLMs. We assume that these code LLMs usually introduce an FIM loss objective in training, which is the same as the testing scenes and greatly improves the repository-level code completion.

**Analysis on different model sizes.** In Table 5, we provide the performance of StarCoder for different model sizes in the validation set of M$^2$RC-EVAL. Notably, StarCoder-7B consistently outperforms StarCoder-3B under comparable conditions. However, following the application of SFT on M$^2$RC-INSTRUCT, the results of StarCoder-3B exceed those of the inference-only StarCoder-7B. This finding underscores the effectiveness of our M$^2$RC-INSTRUCT in augmenting the capabilities of smaller models in repository-level code completion.

Table 5: Performance on M$^2$RC-EVAL.

| Model | Average | |
|---|---|---|
| | EM | ES |
| StarCoder-3B | 14.9 | 43.5 |
| + Retrieval | 14.6 | 38.4 |
| + Retrieval & Tuning | 41.7 | 69.1 |
| StarCoder-7B | 20.6 | 49.9 |
| + Retrieval | 23.6 | 49.3 |
| + Retrieval & Tuning | 44.4 | 71.4 |

**Analysis on different training data sizes.** In Table 6, we evaluate the fine-tuned StarCoder-7B by employing varying sizes of M$^2$RC-INSTRUCT and report the results on the validation set of M$^2$RC-EVAL. Our observations indicate that increasing the dataset from 0.1k to 50k samples per language yields improved results. This suggests that more training data can help boost the model's performance. Therefore, we select 50k samples per language as the default training set size.

**Analysis on the granularity of different bucket levels.** As mentioned in §( 3.4), we categorize M$^2$RC-EVAL into ten bucket levels based on the positions of the code requiring completion within the abstract syntax tree. As shown in Fig. 5, we presents the performance of StarCoder-7B on the test set of M$^2$RC-EVAL across these different bucket levels, and we observe that as the bucket level

---

[5]https://www.anthropic.com/news/claude-3-5-sonnet

Table 6: Performance on $M^2$RC-EVAL using different training data sizes.

| Data Size (Per lang.) | 100 | 1k | 5k | 10k | 50k |
|---|---|---|---|---|---|
| EM (Avg.) | 23.4 | 35.7 | 40.5 | 42.4 | 44.4 |
| ES (Avg.) | 49.1 | 62.9 | 68.2 | 69.4 | 71.4 |

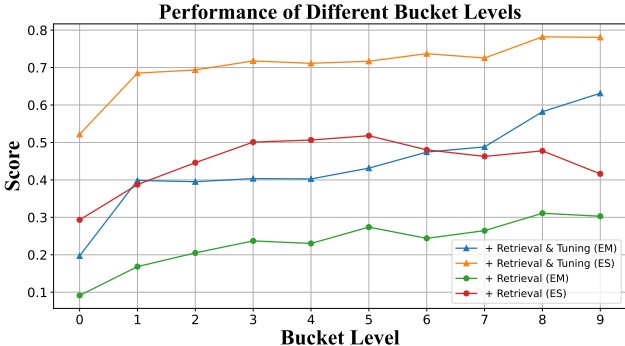

Figure 5: Effectiveness of different bucket levels based on StarCoder-7B.

decreases, the performance of StarCoder-7B correspondingly declines, which means that the code completion on the shadow layer is usually more challenging than on the deep layer. For more experimental data on single-language completion performance and its relation to bucket levels, please refer to Fig.9 and Fig.10 in the Appendix. These findings suggest that the code LLMs encounter challenges when addressing shallow nodes within the syntax tree during the code completion process.

**Analysis on the granularity of different semantic levels.**
Similarly, in §( 3.4), we also categorize the nodes within the abstract syntax tree into eleven primary semantic levels based on their semantic characteristics, and we provide the performance of StarCoder-7B on repository-level code completion for these various semantic levels across multilingual languages on the test set of the $M^2$RC-EVAL. Notably, we observe significant performance disparities across different semantic levels. Specifically, StarCoder-7B shows superior performance on "Identifier and Scope", while it exhibits lower efficacy on "Special Language Structure", This suggests that current code LLMs are proficient at completing tasks related to variable definitions and references, yet their capacity to handle characteristics of different languages requires further enhancement. For single-language completion performance across various node types, please refer to Fig. 11 in the Appendix.

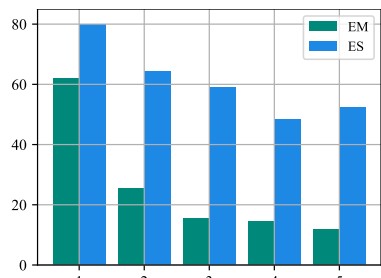

Figure 7: Effectiveness of code completion on different lines based on StarCoder-7B.

**Analysis on completion on different lines.** As shown in Fig.7, StarCoder-7B can effectively complete tasks involving a small number of lines. However, as the number of lines to be completed increases, the scores of the generated code gradually decline. This indicates that completing multi-line code remains a challenge for code LLMs.

Table 8: Performance on $M^2$RC-EVAL.

| Model | Average | |
|---|---|---|
| | EM | ES |
| + Retrieval | 23.6 | 49.3 |
| + Retrieval & Tuning | 44.4 | 71.4 |
| + Retrieval & Tuning (Python Only) | 39.2 | 67.9 |

**Analysis on cross-lingual transfer.**
We fine-tune the StarCoder-7B model using Python-only data (50k) in $M^2$RC-INSTRUCT and compare it with the results of using our whole training data. In Table 8, we report the results on the validation set of $M^2$RC-EVAL, and observe that fine-tuning the model exclusively with Python data

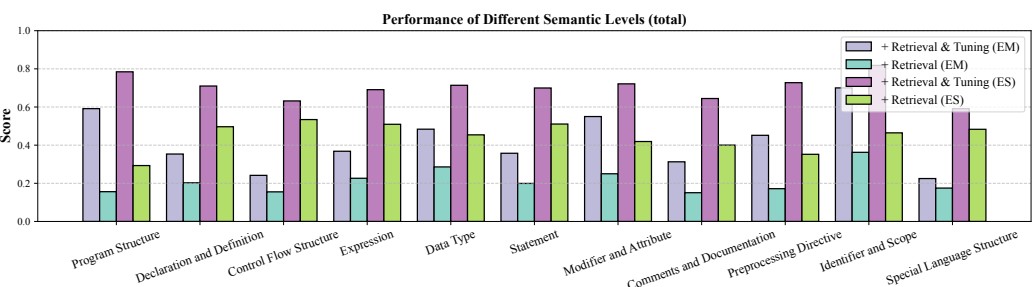

Figure 6: Effectiveness of different semantic levels based on StarCoder-7B.

Table 7: CodeBLEU results on ten representative programming languages.

| Model | C | C# | C++ | Go | Java | JavaScript | PHP | Python | Ruby | Rust | Avg. |
|---|---|---|---|---|---|---|---|---|---|---|---|
| StarCoder-7B | 48.3 | 48.9 | 50.4 | 51.5 | 50.6 | 46.4 | 48.2 | 46.4 | 46.1 | 50.4 | 48.7 |
| + Retrieval | 50.1 | 52.3 | 51.1 | 52.5 | 51.4 | 49.3 | 52.2 | 49.3 | 49.1 | 51.4 | 50.9 |
| + Retrieval & Tuning | 56.0 | 57.4 | 57.6 | 57.0 | 57.6 | 54.8 | 57.8 | 52.0 | 52.9 | 55.5 | **55.9** |

resulted in a significant improvement in its $M^2$RC-EVAL score, coming close to the ES performance achieved through fine-tuning with data from 18 languages. Note that we provide detailed improvements on different languages in Fig. 22 and Fig. 23.

**Analysis on CodeBLEU metric.** In Table 3, we mainly report the EM and ES metrics based on the textual similarity, which neglects important syntactic and semantic features of codes and underestimates different outputs with the same semantic logic. Thus, the CodeBLEU (Ren et al., 2020) [6] is proposed, which considers information from not only the shallow match, but also the syntactic match and the semantic match. In Table 7, we report the results of 10 popular programming languages using the test split of $M^2$RC-EVAL based on the StarCoder-7B model and observe that we can still achieve better performance by fine-tuning on our constructed $M^2$RC-INSTRUCT, which further demonstrates the effectiveness of our $M^2$RC-INSTRUCT on repository-level code completion.

**Analysis on various input lengths.** As shown in Fig. 8, we report the results produced by StarCoder-7B ("Retrieval & Tuning") on our $M^2$RC-EVAL when the input lengths of range in {512, 1024, 2048, 4096} tokens. In Fig. 8, we observe that a scaling law exists, where better performance is achieved when the input length is larger. Thus, we set the default input length as 4096 tokens.

## 5 CONCLUSION

In this paper, we propose the first massively multilingual repository-level code completion benchmark ($M^2$RC-EVAL) with 18 popular programming languages, where two types of fine-grained annotations (bucket-level and semantic-level) are provided to comprehensively analyze the effectiveness of different code LLMs. Besides, we also curate a high-quality instruction corpus $M^2$RC-INSTRUCT to enhance the performance of existing models on repository-level code completion. Extensive experimental results and detailed discussions demonstrate the effectiveness of our proposed $M^2$RC-EVAL and $M^2$RC-INSTRUCT. Finally, we hope $M^2$RC-EVAL could guide the developers and researchers to understand the repository-level code completion capabilities of LLMs and facilitate the growth of code intelligence and software engineering.

Figure 8: Performance on $M^2$RC-EVAL with various input lengths based on StarCoder-7B.

---

[6]We test the CodeBLEU metric based on https://github.com/k4black/codebleu.

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

# A APPENDIX

## A.1 BROADER IMPACTS

In this paper, we propose a repository-level code completion benchmark with 18 programming languages. Therefore, we hope our work can enhance the improvements on the multilingual repository-level code completion task.

## A.2 LIMITATIONS

First, there are several hyperparameters (e.g., training sizes, input length) to tune, which is laborious and expensive. Second, the current work only focuses on the repository-level code completion task, where other repository-level code intelligence tasks are not considered. Third, only textual similarity scores (EM and ES) are used and execution-based evaluation based on test cases is not applied, which may not reflect the performance of different code LLMs well.

## A.3 DETAILS OF THE BASELINE MODELS

**StarCoder** (Li et al., 2023) is a series of generative language models (e.g., 7B, 15.5B). These decoder-only models are trained on the Stack dataset (Kocetkov et al., 2022) and can support 8K tokens in context.

**DeepSeekCoder** (Guo et al., 2024b) is a collection of code-oriented models with capacities from 1.3B to 33B parameters. Trained on a manually curated 2-trillion-token corpus, these models leverage Fill-in-the-Middle (FIM) (Bavarian et al., 2022a) and Rotary Position Embedding (RoPE) (Su et al., 2024) techniques, which enables efficient code generation and infilling within a 16K token window.

**Code Llama** (Roziere et al., 2023) is a family of code large language models based on Llama 2 (Touvron et al., 2023) with 7B, 13B, 34B, and 70B parameters. While trained on 16K token sequences, these models can handle inputs up to 100K tokens during inference.

Note that we just use the base model versions of these three models.

## A.4 DISCUSSION ON NO EXECUTION-BASED EVALUATION

Current datasets for repository-level code completion evaluation, such as CrossCodeEval (Ding et al., 2023) and RepoBench (Liu et al., 2023b), only assess textual similarity between predictions and ground-truth. We hypothesize this limitation stems from several challenges: Firstly, generating comprehensive unit tests for each completion position in a repository is problematic. Single-line completions often fail to construct executable functions, and ensuring adequate test coverage is difficult. Secondly, execution-based evaluation necessitates creating diverse environments for each repository, accommodating various software packages and hardware requirements. This process is intricate and challenging to implement. Thirdly, existing benchmarks with unit tests typically focus on simpler scenarios, like single-file completions or function body generation. Examples include commonly used datasets such as Humaneval CodeGeeX (2022) and MBPP Austin et al. (2021). Despite these obstacles, we recognize the importance of execution-based evaluation for accurately assessing code completion effectiveness, and we will continue to investigate how to evaluate repository-level code completion well.

## A.5 ANALYSIS ON MORE EVALUATION METRICS

In Table 9, for syntax static analysis, following Qwen2.5-Coder (Hui et al., 2024a), to further verify the syntax correctness of the predicted code snippets, we use the code static checking tools (Tree-Sitter) for all predicted code snippets of test split of M2rc-Eval. Specifically, we parse the code snippet into the abstract syntax tree and filter out the code snippet, where the parsed nodes in the code snippet have parsing errors. For execution analysis, as discussed in Appendix A.4, generating unit test cases and providing execution sandboxes for repository-level code completion are very challenging. In this rebuttal phase, we follow RepoCoder (Zhang et al., 2023) to provide the execution test samples in Java language. Specifically, as running tests can be time-consuming

Table 9: Performance on M$^2$RC-EVAL.

| Model | Average | |
|---|---|---|
| | Syntax Accuracy | Execution Accuracy |
| + Retrieval | 86.8 | 48.5 |
| + Retrieval & Tuning | 85.9 | 53.5 |
| + Retrieval & Tuning (Python Only) | 96.9 | 60.4 |

and computationally expensive, we first randomly select a separate set of smaller-scale repositories that are easy to deploy. Besides, as collecting unit tests can be time-consuming, we directly utilize the unit tests available in these repositories and annotate corresponding functions covered by these unit tests. Finally, we utilize unit tests present in the repository to evaluate the functional correctness of the completed function body, where we report the Pass@1(Pass rate is 1 if the code passes all the corresponding test cases, and 0 otherwise). Note that the number of samples for execution analysis is 175, and the average number of test cases is 4.3. Results for syntax static analysis and execution analysis are shown in Table 9, and we observe that both the execution accuracy and syntax accuracy improve a lot after tuning. Notably, the syntax accuracy is close to 100% after tuning, which means that existing code LLMs can easily learn the basic syntax rules for existing programming languages.

A.6 ANALYSIS ON THE QUALITY CONTROL

In §( 3.2), we discard test samples that could be exactly predicted by DeepSeekCoder-1.3B without cross-file contexts. Meanwhile, to discuss more clearly, we also use the DeepSeekCoder-6.7B, StarCoder-7B, and DeepSeekCoder-33B to analyze the ratios of evaluation cases with or without using repository-level contexts. Specifically, we prompt DeepSeekCoder-6.7B, StarCoder-7B, and DeepSeekCoder-33B using the in-file contexts of each sample and obtain three predictions. If one prediction is exactly matched ground-truth, this sample is considered to be predicted without requiring repository-level contexts. Finally, we observe that 71% samples cannot be well predicted only using in-file contexts, which indicates that it is necessary to use the cross-file contexts to achieve better performance in our M$^2$RC-EVAL.

A.7 ANALYSIS ON FAILURE CASES IN DIFFERENT PROGRAMMING LANGUAGES

We manually inspect the behavior of StarCoder-7B (Li et al., 2023) on completion cases in different programming languages. As shown in Fig. 15, the model successively predicts the attribute `position.y`, which is an easy pattern that could be inferred from `position.x` in the prefix. Besides, the `(x, y)` pattern that occurs multiple times in the cross-file context. On the contrary, the model seems to struggle with complex expressions and statements. In Fig. 16, the model should complete the function with a combined condition and return statement. However, the retriever could not provide useful references and the model only predicts half of the condition correctly. Fig. 17 illustrates a Python script to execute memory calculations. Although some calculations appear in the cross-file context, there are no precisely matched calculation procedures. The recurrent conditions in the ground truth require calculations on the data shape, but the model clumsily guesses the data shape. Moreover, we observe that the model prediction is usually affected by frequent identifiers in the retrieved contents. In Fig. 18, the model repeats the `gc.Client` and results in a hallucination for object `PullRequests`, where the ground truth is `gc`. Similarly, in Fig. 19, the model blindly catches the "err" with error level. Yet the correct log level is "warn", which could be judged from `c.on('error', console.warn)` in the Cross file Context 1. Further, in Fig. 20, the ground truth and the model prediction differ by only two characters "()" from a textual perspective, but the ground truth passes the method reference while the model prediction passes the method return.

A.8 ANALYSIS ON LANGUAGE-SPECIFIC INSIGHTS

We have classified 18 programming languages in M$^2$RC-EVAL into 5 programming paradigms and 9 application scenarios as shown in Table 10 and Table 11: Based on the above programming classification structure, we also report the EM results based on StarCoder-7B as shown in Table 12

and Table 13, and have the following observations: (1). For different programming paradigms, we observe that the markup language paradigm has the lowest performance, and the functional paradigm has the best performance. Besides, after tuning, the performance of the markup language paradigm improves greatly. We assume that the syntax rules for markup language are easy, and these code LLMs can quickly obtain these rules after tuning. (2). For different application scenarios, the performance varies significantly. Specifically, the Web Frontend and Scientific Computing have relatively low performance, which needs to be improved for existing code LLMs. (3). We observe these LLMs share some common strengths and weaknesses and we will continue to investigate more language-specific insights to better improve the code completion abilities of existing code LLMs.

Table 10: Classification of M$^2$RC-EVAL based on paradigm types.

| Paradigm Types | Languages |
|---|---|
| Procedural | C |
| Object Oriented | C#, Java, Kotlin, Objective-C |
| Multiple Paradigms | C++, Go, JavaScript, Lua, PHP, Python, R, Ruby, Rust, Scala, TypeScript |
| Functional | Haskell |
| Markup Language | HTML |

Table 11: Classification of M$^2$RC-EVAL based on application scenarios.

| Application Scenarios | Languages |
|---|---|
| Mobile | Kotlin, Objective-C |
| Cross Platform | Java |
| Desktop Application | C# |
| Web Frontend | JavaScript, TypeScript, HTML |
| Web Backend | Go, PHP, Ruby, Rust, Scala |
| Scientific Computing | Python, R |
| System & Software | C, C++ |
| Education & Research | Haskell |
| Automation Scripts | Lua |

Table 12: Results of M$^2$RC-EVAL based on paradigm types.

| Paradigm Types | StarCoder | + Retrieval | + Retrieval & Tuning |
|---|---|---|---|
| Procedural | 19.2 | 23.7 | 47.6 |
| Object Oriented | 21.3 | 24.2 | 47.4 |
| Multiple Paradigms | 21.1 | 24.3 | 43.4 |
| Functional | 25.1 | 24.8 | 44.6 |
| Markup Language | 17.2 | 21.3 | 46.7 |

## A.9 ANALYSIS ON THE COMPLETION CURSOR POSITION

As mentioned in many works (Hui et al., 2024b; Jiang et al., 2024), developers often expect LLMs to complete the current code into a complete snippet, such as a completed code line or loop block, instead of suggesting an incomplete code snippet. Besides, the recent Qwen2.5-Coder adopts a similar way with M$^2$RC-INSTRUCT to produce the instruction dataset. Meanwhile, to demonstrate the effectiveness of our M$^2$RC-INSTRUCT, we also constructed a test dataset called M$^2$RC-EVAL (Random). Specifically, for a fair comparison, based on the same repositories of M$^2$RC-EVAL, we randomly choose arbitrary completion cursor positions while ensuring the completed code forms a valid AST node, where the generated testing set is named M$^2$RC-EVAL (Random). The evaluation results are as shown in Table 14. We observe that higher performance is obtained in M$^2$RC-EVAL (Random). For this phenomenon, the possible reason is as follows. In each completion span, the completion cursor position of our default M2rc-Eval is the start position of the syntax node block,

Table 13: Results of M$^2$RC-EVAL based on application scenarios.

| Application Scenarios | StarCoder | + Retrieval | + Retrieval & Tuning |
|---|---|---|---|
| Mobile | 21.2 | 22.9 | 48.0 |
| Cross Platform | 24.2 | 25.3 | 48.1 |
| Desktop Application | 18.7 | 26.1 | 45.3 |
| Web Frontend | 17.9 | 22.2 | 41.5 |
| Web Backend | 22.8 | 24.8 | 44.6 |
| Scientific Computing | 18.2 | 23.5 | 40.3 |
| System & Software | 21.3 | 24.0 | 50.1 |
| Education & Research | 25.1 | 24.8 | 44.6 |
| Automation Scripts | 21.7 | 26.3 | 43.7 |

Table 14: Results of M$^2$RC-EVAL and M$^2$RC-EVAL (Random).

| Model | EM/ES (M$^2$RC-EVAL) | EM/ES (M$^2$RC-EVAL (Random)) |
|---|---|---|
| StarCoder-7B | 21.0/52.0 | 24.0/53.9 |
| + Retrieval | 24.1/50.0 | 30.1/55.0 |
| + Retrieval & Tuning | 44.5/72.2 | 54.5/76.8 |
| DeepSeekCoder-6.7B | 22.6/54.7 | 25.3/56.2 |
| + Retrieval | 25.1/51.7 | 32.4/59.6 |
| + Retrieval& Tuning | 46.8/74.1 | 55.7/78.3 |

which means that no context can be used inside the syntax node block. In contrast, the completion cursor position of M$^2$RC-EVAL (Random) is the arbitrary position of the syntax node block, which means that there exist additional informative contexts inside the current syntax node block for better completion. In other words, in M$^2$RC-EVAL (Random), the additional contexts are inside the syntax node block and before the completion cursor position, which can decrease the completion difficulty and improve the completion quality.

## A.10 MORE EXPERIMENTS

- We provide the analysis on the bucket levels in Fig. 9 and Fig. 10, respectively.
- We analyze the effect of different semantic levels on Rust, Objective-C, and Haskell in Fig. 11. respectively.
- We provide the semantic-level annotations on 18 languages in Fig. 12, Fig. 13 and Fig. 14.
- We provide the results of problems from different difficulty levels in Fig. 21, where we define completion on 1 line, completion on 2-3 lines and completion on 4-5 lines as easy, middle, and hard settings, respectively.
- The prompt template for evaluating the repository-level code completion of general LLMs is shown in Template A.10.

---

**Prompt Template**

**System Instruction:**

## First, input a segment of <LANG> code that needs completion. Please help complete the code at the corresponding position.

## The format of the input code is as follows:
<fim_start> prefix <fim_hole> suffix <fim_end>

Explanation:
1. <fim_start>, <fim_hole>, and <fim_end> are special characters.
2. <fim_hole> is the position that needs completion.
3. The prefix after <fim_start> represents the context before the content that needs completion.
4. The suffix after <fim_hole> represents the context following the content that needs completion.

## The output format is as follows:
    1. Only the code completion result for the position <fim_hole> is needed.
    2. Do not use markdown format.
    3. Do not include the surrounding context.
    4. Do not provide any explanation or description.

## The content of the input code is as follows:


---

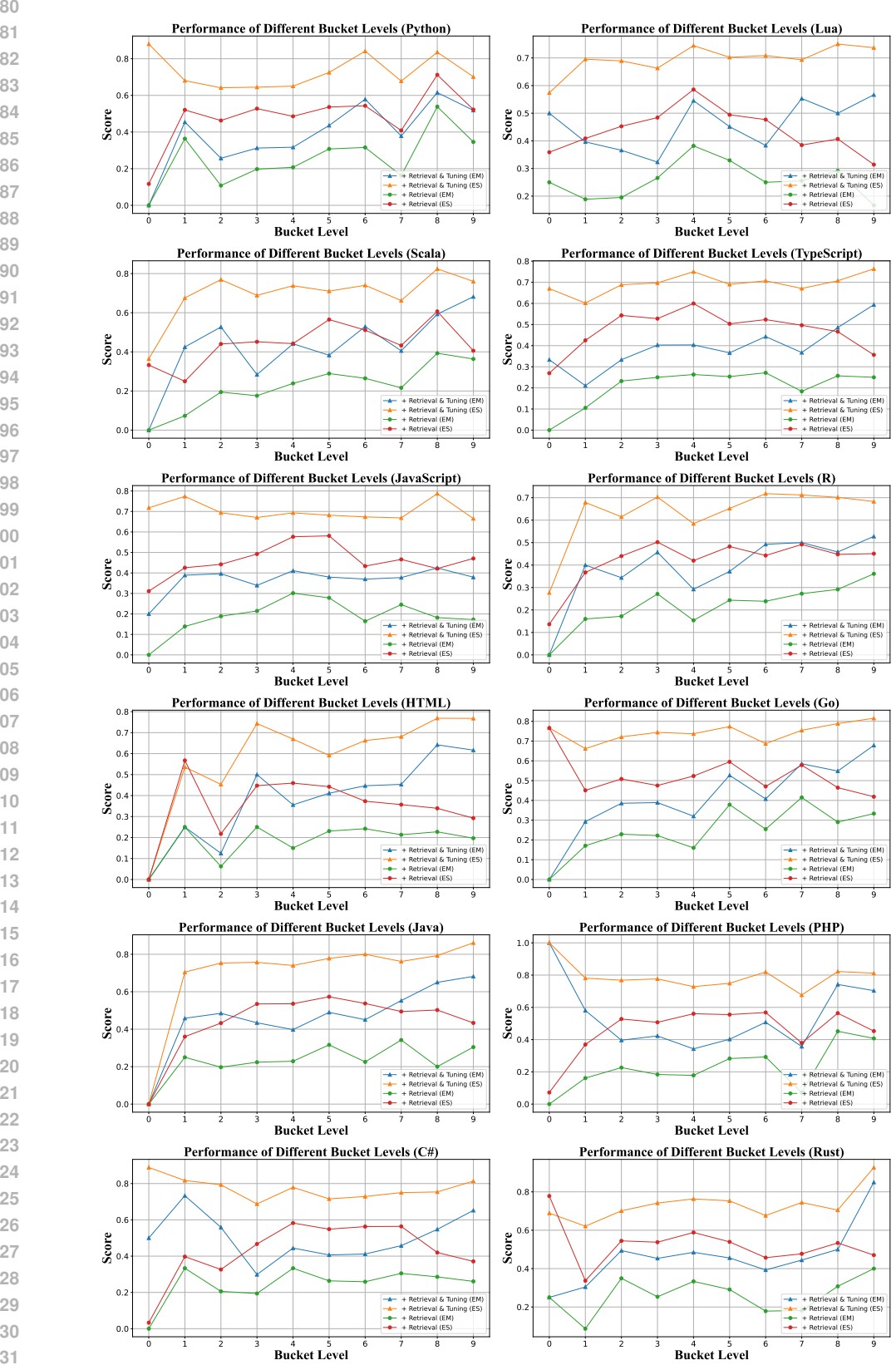

Figure 9: Effectiveness of different bucket levels based on StarCoder-7B for different languages.

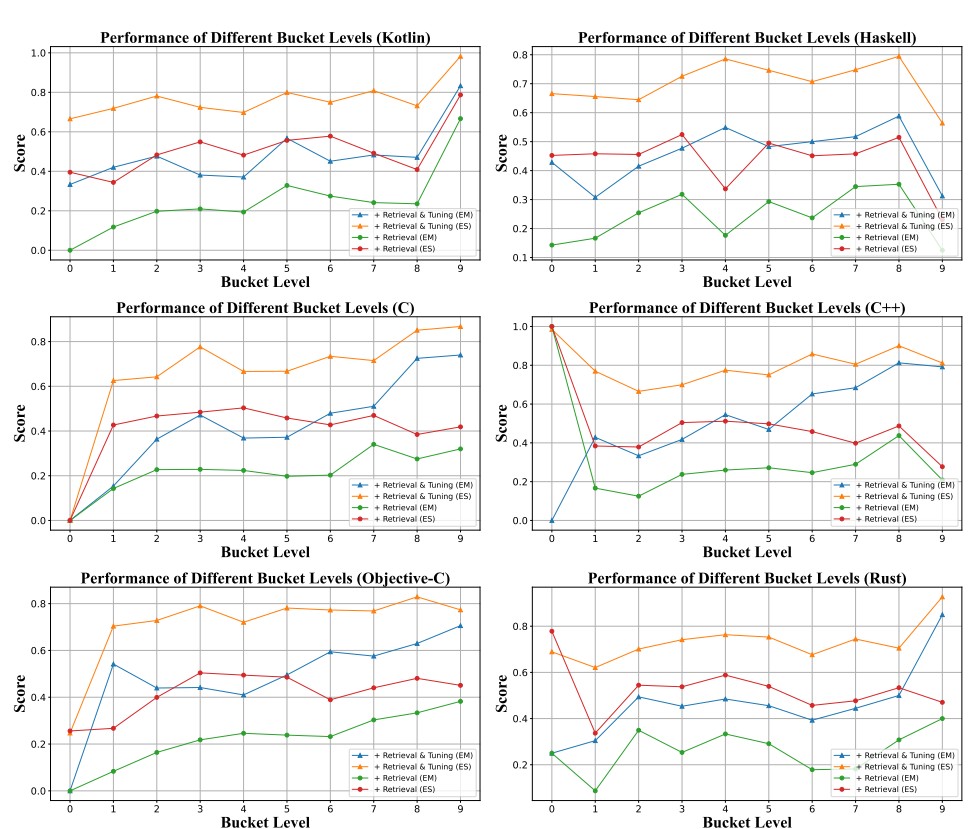

Figure 10: Effectiveness of different bucket levels based on StarCoder-7B for different languages.

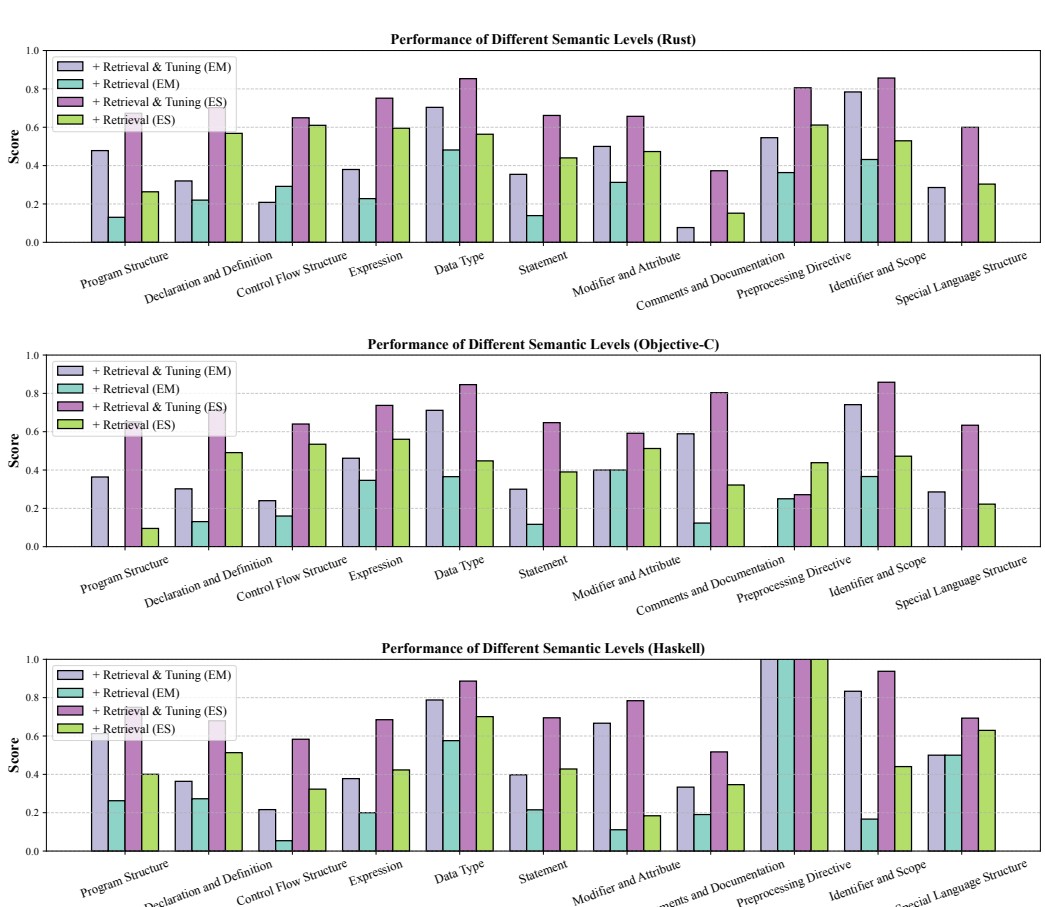

Figure 11: Effectiveness of different semantic levels based on StarCoder-7B.

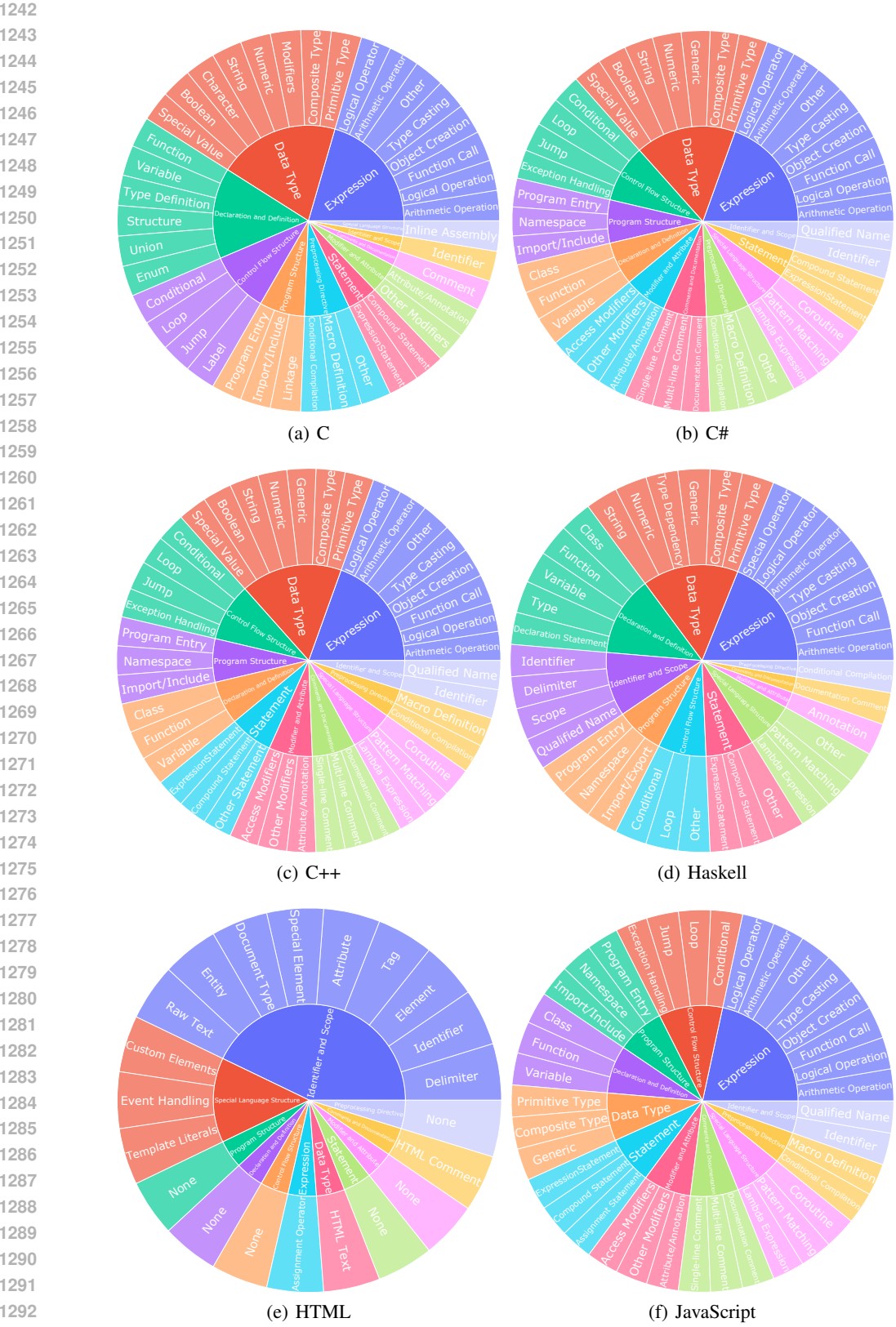

Figure 12: Semantic-level annotations on different types of programming languages. "none" is used if this language does not have corresponding subcategories.

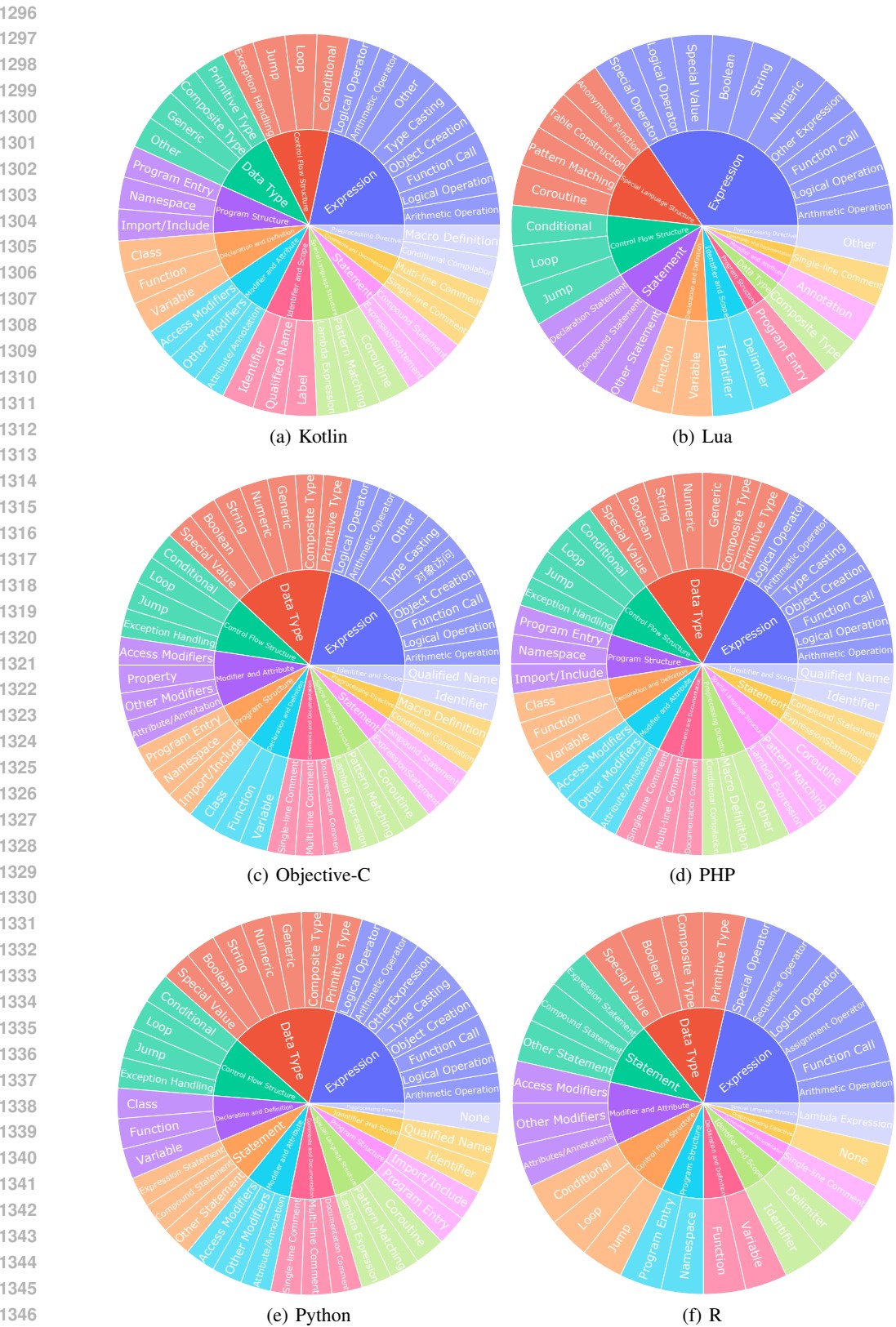

Figure 13: Semantic-level annotations on different types of programming languages. "none" is used if this language does not have corresponding subcategories.

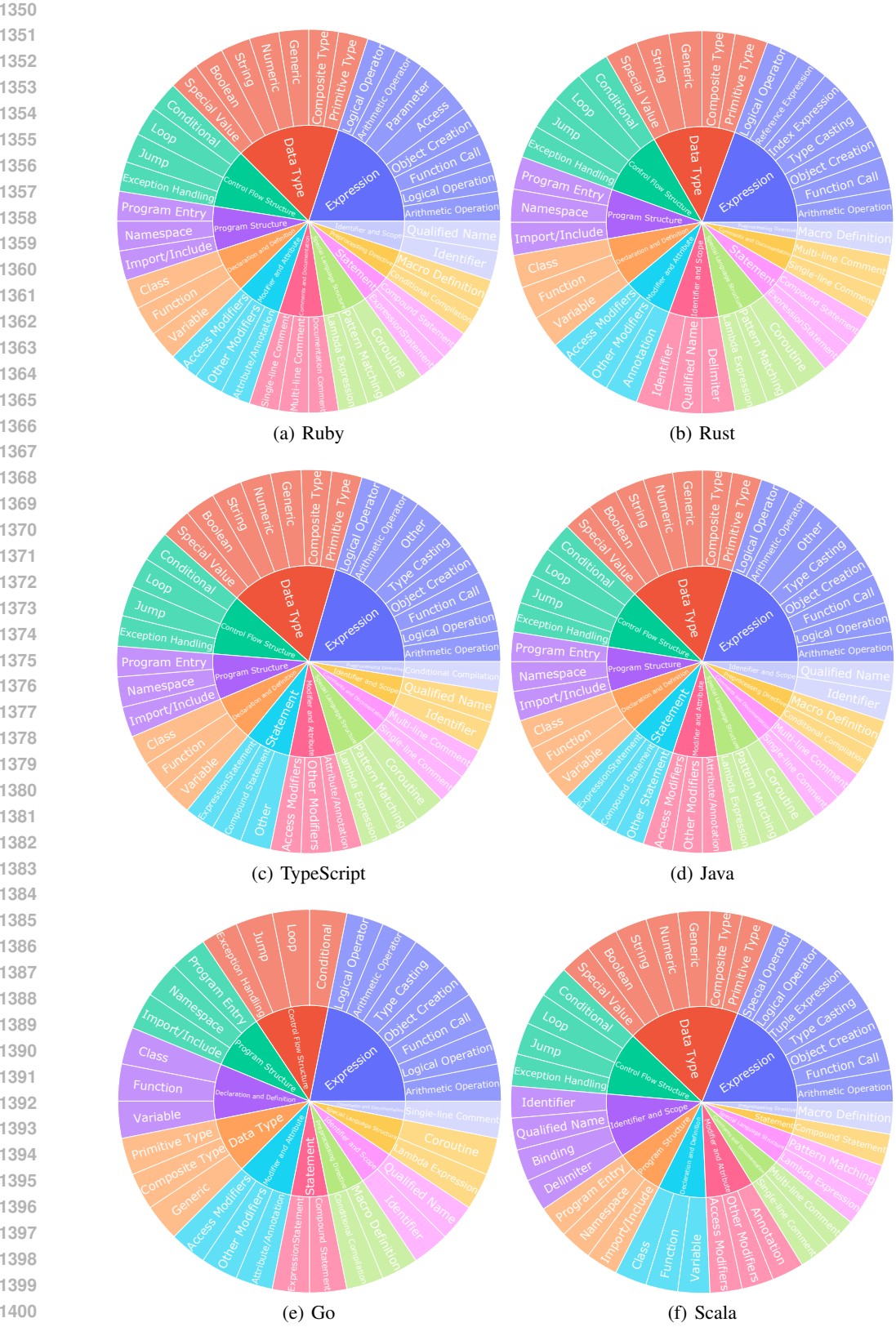

Figure 14: Semantic-level annotations on different types of programming languages. "none" is used if this language does not have corresponding subcategories.

```typescript
// Path: /src/ts/Enemy/Enemy.ts                    Cross file Context 1
protected setPosition({ x, y }: Position): void {
  this.position = { x, y };
}
public getCurrentTile(): Tile {
  return map.getTile(this.position);
}
protected setDestinyTile(tile: Tile): void {
  this.destinyTile = tile;
}
getDestinyTile(): Tile {
```

```typescript
// Path: /src/ts/Enemy/Enemy.ts                    Cross file Context 2
}
public setEnemyFree(): void {
  this.isFree = true;
  this.setGameMode( GameMode.CHASE);
}
protected setPosition({ x, y }: Position): void {
  this.position = { x, y };
}
public getCurrentTile(): Tile {
  return map.getTile(this.position);
```

```typescript
// Path: /src/ts/Enemy/Enemy.ts                    Cross file Context 3
tweenMovement(image, () => {
  image.destroy();
  this.ghost.x = CENTER_MAP_POSITION.x;
  this.ghost.y = CENTER_MAP_POSITION.y;
  enemySprite.body.moves = true;
  enemySprite.visible = true;
  this.ghost.anims.play(`ghost${this.ghostType}East`);
});
setTimeout(() => {
  enemySprite.enableBody();
```

```typescript
// Path: /src/ts/Enemy/Enemy.ts                    Cross file Context 4
Compare with this code snippet:
        this.ghost.y += this.SPEED;
        break;
    case "NORTH":
        animationName = "North";
        this.ghost.y -= this.SPEED;
        break;
    case "WEST":
        animationName = "West";
        this.ghost.x -= this.SPEED;
```
••••••

```typescript
                                                  In-file Context
import { Tile } from '../Tile';
import { map, pacman, ENEMY_SPAWN_TIME } from '../app'
import { GameMode } from '../game-interfaces/modes.interface';
import { scene } from '../app'
import { Utils } from '../Utils/utils';

export class RedGhost extends Enemy {
    private scatterPosition

    constructor( ){
        let position = { x: 475, y: 375 }
        let ghost = scene.physics.add.sprite( position.x,
```
        **Completion Cursor Position**
```typescript
                ,"ghostRedAnim" )
        ghost.type = "Red"
        ghost.timeToSetFree = ENEMY_SPAWN_TIME
        scene.enemyGroup.add(ghost);
        super( position, ghost )
        this.initialPosition = position
        this.scatterPosition = {x:2,y:2}
    }
    private findDestinyTile(): Tile{

        switch( this.mode ){
            case GameMode.CHASE:
                return map.getTile( pacman.getCurrentPosition() ) )
            case GameMode.FRIGHTENED:
                return this.frightenedTile
            case GameMode.SCATTER:
                return map.getTile( this.scatterPosition, 'index' )
        }
    }
}
```

**Ground Truth**
```
position.y
```

🤖 **Model Output** ✅
```
position.y
```

Figure 15: Visualization on success case for TypeScript. (Semantic label: *Modifier and Attribute*)

```c
// Path: /ketopt.h                                 Cross file Context 1
static void ketopt_permute(char *argv[], int j, int n)
{
    int k;
    char *p = argv[j];
    for (k = 0; k < n; ++k)
        argv[j - k] = argv[j - k - 1];
    argv[j - k] = p;
}
```

```c
// Path: /Correct.h                                Cross file Context 2
inline int calculate_score(int new_occ_0, int new_occ_1)
{
    if(new_occ_0 + new_occ_1 == 0)
    {
        return -1;
    }
    if(filter_snp(new_occ_0, new_occ_1, new_occ_0 + new_occ_1) == 0)
    {
        return -1;
    }
```

```c
// Path: /Levenshtein_distance.h                   Cross file Context 3
        (*return_err) = line_error;
    }
    return (*return_t_end);
}
inline void reverse_string(char* str, int strLen)
{
    int i, Len;
    char k;
    Len = strLen / 2;
    for (i = 0; i < Len; i++)
```

```c
// Path: /Correct.h                                Cross file Context 4
    double threshold = 0.30;
    available = available/((double)(total));
    if(available <= threshold && available < 6)
    {
        return 0;
    }
    return 1;
}
inline int filter_one_snp(int occ_0, int occ_1, int total)
{
```
••••••

```c
#include <pthread.h>                               In-file Context
#include <stdint.h>

typedef struct{
    int *l, n;
    char **a;
}enzyme;

typedef struct {
    int flag;
    int num_reads;
    ......

    uint32_t ul_min_base;
} hifiasm_opt_t;

extern hifiasm_opt_t asm_opt;

void init_opt(hifiasm_opt_t* asm_opt);
void destory_opt(hifiasm_opt_t* asm_opt);
void ha_opt_reset_to_round(hifiasm_opt_t* asm_opt, int round);
void ha_opt_update_cov(hifiasm_opt_t *opt, int hom_cov);
void ha_opt_update_cov_min(hifiasm_opt_t *opt, int hom_cov, int min_chain);
int CommandLine_process(int argc, char *argv[], hifiasm_opt_t* asm_opt);
double Get_T(void);

static inline int ha_opt_triobin(const hifiasm_opt_t *opt)
```
        **Completion Cursor Position**
```c
static inline int ha_opt_hic(const hifiasm_opt_t *opt)
{
    return ((opt->hic_reads[0] && opt->hic_reads[1]));
}
```

**Ground Truth**
```
{return ((opt->fn_bin_yak[0] && opt->fn_bin_yak[1])
       || (opt->fn_bin_list[0] && opt->fn_bin_list[1]));}
```

🤖 **Model Output** ❌
```
{return (opt->fn_bin_poy && opt->fn_bin_yak[0] && opt->fn_bin_yak[1]);}
```

Figure 16: Visualization on failure case for the C language. (Semantic label: *Statement*)

Figure 17: Visualization on failure case for Python. (Semantic label: *Expression*)

Figure 18: Visualization on failure case for Go. (Semantic label: *Expression*)

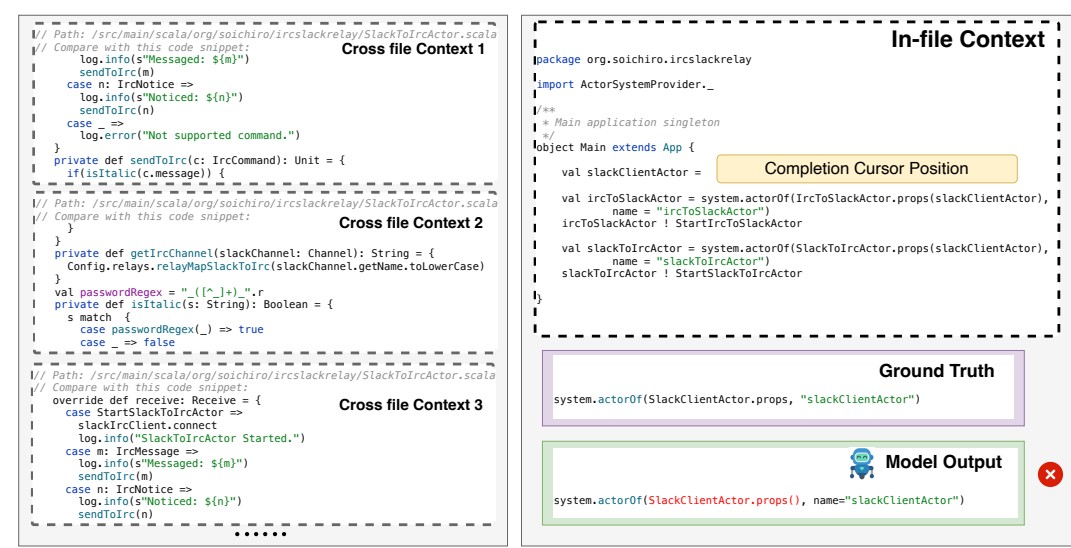

Figure 19: Visualization on failure case for Javascript. (Semantic label: *Statement*)

Figure 20: Visualization on failure case for Scala. (Semantic label: *Expression*)

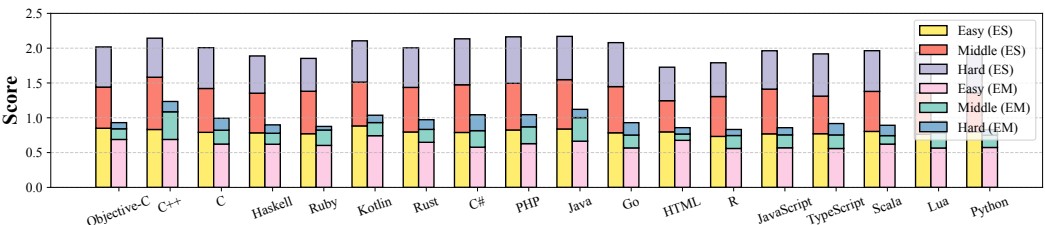

Figure 21: Performance on M²RC-EVAL for problems of different difficulty levels.

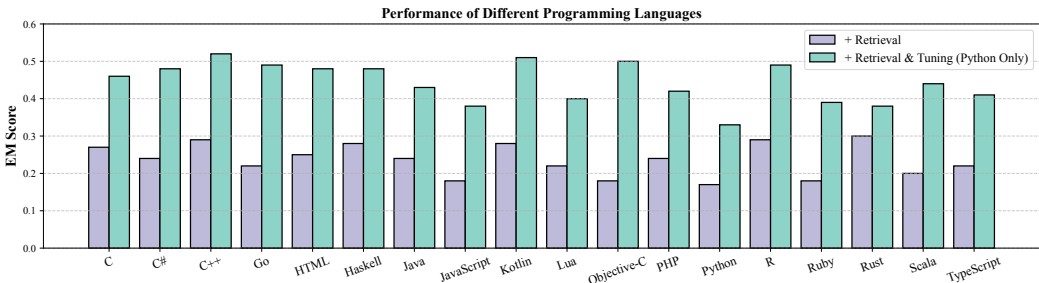

Figure 22: EM performance on $M^2$RC-EVAL for different programming languages when only using Python training data.

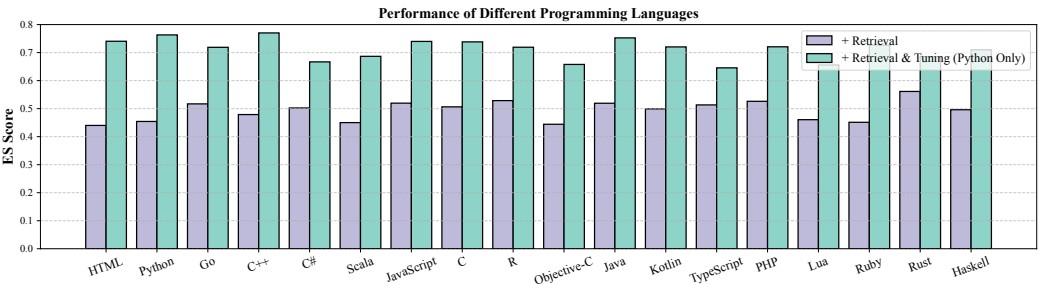

Figure 23: ES performance on $M^2$RC-EVAL for different programming languages when only using Python training data.

