# OpenReview forum: "M2rc-Eval: Massively Multilingual Repository-level Code Completion Evaluation"
_ICLR.cc/2025/Conference — Submitted to ICLR 2025_

### Official Review · Reviewer_5s7R · 2024-10-28

**Soundness:** 3
**Presentation:** 4
**Contribution:** 4
**Rating:** 8
**Confidence:** 3

**Summary:**

This paper introduces M2RC-EVAL, a new benchmark designed to assess repo-level code completion performance across 18 programming languages. The benchmark uses two types of fine-grained annotations: bucket-level, based on the depth of abstract syntax trees, and semantic-level, which focuses on code semantics.  Additionally, the paper presents a multilingual instruction corpus, M2RC-INSTRUCT, to improve code completion models.

**Strengths:**

**Massive multilingual benchmark**: The paper introduces a novel multilingual benchmark that supports 18 programming languages, which is a significant expansion over existing benchmarks that only handle a limited number of languages.

**Quality control**: The dataset construction is considered thorough, with a robust collection process from github repos and clear quality control measures.

**Clear writings**: This paper is well-organized, with clear explanations of how the benchmark and instruction dataset were constructed and evaluated. The use of tables and figures aids in understanding the experimental results.

Overall, this benchmark contributes to address the growing need for multilingual evaluation in code completion. The fine-grained annotation methodology also provides useful insights into the challenges faced by models in many programming scenarios.

**Weaknesses:**

**Evaluation Metrics:** While the paper mainly uses exact match and edit similarity metrics, these may not fully capture the functional correctness of the generated code. I acknowledge that implementing an execution-based evaluation across such a massive multilingual benchmark poses considerable challenges, and this is more of a broader limitation within the field rather than a direct weakness of this particular work. I do appreciate the authors’ inclusion of CodeBLEU, which offers another perspective by considering syntactic and semantic aspects of code generation. Still, it would be more insightful to explore whether there is any correlation between CodeBLEU scores and the more traditional EM and ES metrics.

**Questions:**

Apart from the point in the 'Weaknesses' section, I have the following questions and suggestions:

**Proprietary Model Comparison**: Do the authors plan to include a comparison with advanced proprietary models, even on a smaller subset of the benchmark? This could offer insights into how fine-tuned open-source models compare against the leading commercial solutions.

**Qualitative Error Analysis (Suggestion):** It would be insightful to provide a more detailed qualitative error analysis to highlight where current code LLMs encounter the most difficulty across different languages. This could help pinpoint areas for further improvement.

**Figure Clarity (Suggestion):** While the visualizations are nicely presented, some figures (e.g., Figures 4 and 14) have very small fonts, making it difficult to read the details. Increasing the font size in these visualizations would enhance clarity and accessibility for readers.

---

> ### Author Response · Authors · 2024-11-24
>
> **Q1: Evaluation metrics.**
>
> **A1**: Thanks for your positive feedback on our paper. We have additionally provided more evaluation results on General Response (G.Q2).
>
>
>
> **Q2: Correlation between CodeBLEU scores and the more traditional EM and ES metrics.**
>
> **A2**: Firstly, the definitions of EM and ES are as follows: Exact Match (EM) evaluates the percentage of cases where the prediction exactly matches the reference, providing a strict measure of how often LLMs produce correct code without deviations. Edit Similarity (ES) measures the similarity between the prediction and the reference based on the number of edits required to transform one into the other, typically using metrics like Levenshtein distance.
>
> However, EM and ES metrics only assess the textural similarities and ignore semantic structure similarities among predictions and ground truth. Specifically, the limitations of EM and ES are as follows: (1) The perfect accuracy of EM is too strict, and underestimates different outputs with the same semantic logic. (2) ES score does not take into account the grammatical and logical correctness, resulting in favoring candidates with high ES accuracy and serious logical errors.
>
> To address these limitations, the CodeBLEU is proposed, which considers information from the shallow (n-gram) match, the syntactic match, and the semantic match. More specifically, the n-gram match assigns different weights for different n-grams, the syntactic match considers the abstract syntax tree (AST) information in the evaluation score by matching the sub-trees, and the semantic match uses data-flow structure to measure the semantic similarity.
>
> To analyze the correlation between CodeBLEU scores and the more traditional EM and ES metrics, we analyze the **EM/ES/CodeBLEU** scores for six models using retrieval in the following table, where we define **completion on 1 line, completion on 2-3 lines and completion on 4-5 lines as easy, middle, and hard settings**, respectively.
>
> The benchmark results (EM/ES/CodeBLEU) are as follows:
>
> | **Model**          | **Easy**       | **Medium**     | **Hard**       |
> | ------------------ | -------------- | -------------- | -------------- |
> | Code Llama-7B      | 26.7/42.4/46.0 | 13.4/47.3/53.4 | 12.8/53.6/57.1 |
> | StarCoder-7B       | 32.2/47.6/47.0 | 10.0/52.4/54.0 | 13.3/53.8/56.8 |
> | DeepSeekCoder-6.7B | 33.0/49.0/47.7 | 17.1/54.9/58.5 | 9.7/57.0/55.8  |
> | DeepSeekCoder-33B  | 34.9/50.2/48.2 | 17.0/42.8/51.0 | 10.4/46.7/51.0 |
> | Qwen2.5-Coder-7B   | 34.0/49.3/47.9 | 19.9/54.7/59.0 | 12.6/56.9/57.2 |
> | Qwen2.5-Coder-32B  | 53.6/69.0/53.2 | 26.0/62.2/61.2 | 16.0/57.9/57.6 |
>
> Then, we compute  **Spearman's Rank correlation coefficient as the ranking relationships between CodeBLEU and EM/ES scores**, where a larger value denotes greater correlation. The Spearman's Rank correlation coefficient results are as follows.
>
> |                 | **Easy** | **Medium** | **Hard**  |
> | --------------- | ---- | ------ | ----- |
> | CodeBLEU v.s.EM | 1.0  | 0.771  | 0.714 |
> | CodeBLEU v.s.ES | 1.0  | 0.943  | 0.600 |
>
> We observe that  Spearman's Rank correlation coefficient is lower when increasing the difficulty (completion line), which is also reasonable. The possible reason is as follows.
>
> The completion span with larger completion lines usually includes more complex code syntax and semantic structure information, which is considered in the CodeBLEU metric. However, the EM/ES does not consider this structure information.
>
>
>
>
>
> **Q3: Proprietary model comparison.**
>
> **A3**: We have provided a detailed comparison with recent powerful proprietary models in **General Response (G.Q1)**. Please refer to General Response (G.Q1) for more details.
>
>
>
> **Q4: Qualitative error analysis.**
>
> **A4**: Thanks for your insightful advice.  We have provided a detailed qualitative error analysis in Appendix A.7 in our new version.
>
> **Q5: Figure Clarity.**
>
> **A5**: Thanks for your suggestions. For Figure 4  in the original version, we have replaced the figure with a table for better illustration. For Figure 14 in the original version, we have greatly increased the font size and improved the illustration in Figure 12, Figure 13, and  Figure 14.

---

> > ### Comment · Reviewer_5s7R · 2024-11-26
> > **Response to Rebuttal**
> >
> > Thanks for the detailed response. The response clearly addressed my concerns. As my score is quite positive already, I will maintain my score.

---

### Official Review · Reviewer_PyQh · 2024-11-03

**Soundness:** 3
**Presentation:** 3
**Contribution:** 3
**Rating:** 6
**Confidence:** 4

**Summary:**

The paper releases a cross-file-context repo-level code completion benchmark that tests the fill-in-the-middle (FIM) completion abilities of Code LMs. GitHub repos are collected from Stack V2, and then the files' ASTs are individually processed to create varying categories of FIM holes for completion.

I believe the dataset (especially the instruct split) is a valuable OS contribution. Still, as it joins a whole host of existing benchmarks in this area, it should be fleshed out further regarding research questions.

**Strengths:**

1. The paper enters a crowded field of repo-level code completion benchmarks but differentiates itself from the pack by the size of the data splits and the number of languages supported.
2. The data collection process is clearly detailed

**Weaknesses:**

1. For a benchmark paper, it is a bit light on details of env setup, tooling and sandboxing
2. The evaluation is still based on surface form metrics like exact-match, which can be very noisy, especially for FIM settings. Scaling execution tests can be hard, but existing benchmarks like EvoCodeBench [1] and BigCodeBench [2] have done so using a combination of model generation and human validation.
3. Several very strong recent code model releases, such as Yi-Coder [3], Llama-3.1 [4], and Qwen-2.5-coder [5], are missing from the evals. This is pertinent due to their longer context support (up to 128k tokens), allowing for a much more interesting baseline than the paper currently uses (just the individual file in question). A very interesting evaluation of the paper is missing: how well the benchmark can be solved if the model just placed all the most pertinent files in context instead of just retrieving a few short snippets. Similarly, how would the various in-repo file ordering schemes previously explored by StarCoder-2 [6] and DeepSeek-Coder [7] affect this?

[1] https://arxiv.org/abs/2404.00599
[2] https://arxiv.org/abs/2406.15877
[3] https://arxiv.org/abs/2403.04652
[4] https://arxiv.org/abs/2407.21783
[5] https://arxiv.org/abs/2409.12186
[6] https://arxiv.org/abs/2402.19173
[7] https://arxiv.org/abs/2401.14196

**Questions:**

1. The cross-lingual transfer benchmarks may not be properly baseline. The paper uses StarCoder-1, a multi-PL pre-trained model that is continually pre-trained on Python code. This is a rather non-standard setting and would be fairer (across langs) if it were started from StarCoder-1-Base.
2. It would be interesting to explore the effect of various code retrievers instead of just Jaccard-Similarity in retrieving snippets in-context.

---

> ### Author Response · Authors · 2024-11-24
>
> **Q1: Details of env setup, tooling, and sandboxing.**
>
> **A1**: Thanks for your suggestions. In our main paper, we mainly use the EM, ES, and CodeBLEU metrics to evaluate the performance of different LLMs, where EM and ES mainly assess the textual similarities and the CodeBLEU additionally considers the semantic structure. Besides, computing these metrics does not rely on any execution sandbox and we just follow repositories of CrossCodeEval [https://github.com/amazon-science/cceval] and CodeBLEU [https://github.com/k4black/codebleu] to obtain these metrics. We have provided these details in our new version.
>
> **Q2: The evaluation is still based on surface form metrics like exact match. Scaling execution tests can be hard.**
>
> **A2**: Please See **General Response (G.Q2)**.
>
> **Q3: Evaluation of more strong recent models.**
>
> **A3**: Please See **General Response (G.Q1)**. In general response, we have provided more evaluation results of different LLMs in our new version. Note that as Yi-Coder [https://huggingface.co/01-ai/Yi-Coder-9B/blob/main/tokenizer_config.json] has not supported the FIM, we have not provided the results of Yi-Coder.
>
> **Q4: How well the benchmark can be solved if the model just placed all the most pertinent files in context instead of just retrieving a few short snippets?**
>
> **A4**: Thanks for your suggestions. We take the Qwen2.5-Coder-32B with 32K length as an example to test the results in the validation set of M2rc-Eval with longer input as follows. Specifically, for Qwen2.5-Coder-32B (Infile), we only use the infile context for code completion. For Qwen2.5-Coder-32B (Snippet Retrieval, 4K), we follow the same retrieval setting by retrieving relevant code snippets to construct a completion prompt with 4096 tokens. For Qwen2.5-Coder-32B (File-wise Retrieval, Topological, 4K), we just use the relevant pertinent files based on the similarity scores and order these files based on the topological sort algorithm in DeepSeekCoder. For Qwen2.5-Coder-32B (File-wise Retrieval, Random, 4K), we directly randomly order the relevant files to construct the prompt with 4096 tokens.
>
> Similarly, we also report the results of 8K, 16K, and 32K lengths and we have the following observations.
>
> (1). First, when increasing the completion prompt length, lower performance results are obtained for all retrieval methods, which means that the Qwen2.5-Coder-32B cannot handle well in long input although the input length is less than 32K. This phenomenon is similar to the conclusion of many general long-context understanding methods (e.g., Ruler[1], LV-Eval[2], STRING[3] ), which claim that the really effective length is less than the claimed length a lot.
>
> (2). Second, the default snippet retrieval achieves the best performance in 4K length. We assume that the file-wise retrieval will include many noisy or redundant contexts, which introduces more challenges to obtaining the informative contexts for repository-level code completion.
>
> (3). Third, the ordering method based on the topological sort is relatively better than the random baseline in the file-level retrieval setting, which indicates that it is effective to use Topological ordering and it may be a good choice to investigate a more suitable ordering strategy for these retrieved code snippets.
>
> |                                                           | EM     | ES     | CodeBLEU |
> | --------------------------------------------------------- | ------ | ------ | -------- |
> | Qwen2.5-Coder-32B (Infile)                                | 0.3456 | 0.6442 | 0.5374   |
> | Qwen2.5-Coder-32B (Snippet Retrieval, 4K)                 | 0.4122 | 0.6721 | 0.5693   |
> | Qwen2.5-Coder-32B (File-wise Retrieval,  Topological, 4K) | 0.3960 | 0.6695 | 0.5632   |
> | Qwen2.5-Coder-32B (File-wise Retrieval, Random, 4K)       | 0.3872 | 0.6658 | 0.5601   |
> | Qwen2.5-Coder-32B (Snippet Retrieval, 8K)                 | 0.2428 | 0.4732 | 0.5237   |
> | Qwen2.5-Coder-32B (File-wise Retrieval, Topological, 8K)  | 0.2261 | 0.4533 | 0.5180   |
> | Qwen2.5-Coder-32B (File-wise Retrieval, Random, 8K)       | 0.2156 | 0.4424 | 0.5054   |
> | Qwen2.5-Coder-32B (Snippet Retrieval, 16K)                | 0.2006 | 0.4161 | 0.5080   |
> | Qwen2.5-Coder-32B (File-wise Retrieval, Topological, 16K) | 0.1906 | 0.4171 | 0.5061   |
> | Qwen2.5-Coder-32B (File-wise Retrieval, Random, 16K)      | 0.1762 | 0.3977 | 0.4880   |
> | Qwen2.5-Coder-32B (Snippet Retrieval, 32K)                | 0.1783 | 0.3996 | 0.5028   |
> | Qwen2.5-Coder-32B (File-wise Retrieval, Topological, 32K) | 0.1756 | 0.4041 | 0.4988   |
> | Qwen2.5-Coder-32B (File-wise Retrieval, Random, 32K)      | 0.1500 | 0.3747 | 0.4816   |
>
> [1]. RULER: What's the Real Context Size of Your Long-Context Language Models?
>
> [2]. LV-Eval: A Balanced Long-Context Benchmark with 5 Length Levels Up to 256K.
>
> [3]. Why Does the Effective Context Length of LLMs Fall Short?

---

> ### Author Response · Authors · 2024-11-24
>
> **Q5: The cross-lingual transfer benchmarks may not be properly baseline. The paper uses StarCoder-1, a multi-PL pre-trained model that is continually pre-trained on Python code. This is a rather non-standard setting and would be fairer (across langs) if it were started from StarCoder-1-Base.**
>
> **A5**: Sorry for this misleading. It should be mentioned that we just use the base model of StarCoder, which is not continued pretrained on Python code. We have clarified this detail of Appendix A.3 in our new version.
>
>
>
> **Q6: Explore the effect of various code retrievers.**
>
> **A6**: To find the most suitable retriever,  we evaluate the code match **(EM/ES)** performance on the validation set with Jaccard similarity, BM25, and UniXCoder in the following Table. Specifically, the UniXcoder is a neural retriever, which costs more response time for inference. In contrast, both Jaccard and BM25 are lexical retrievers, which can reduce infer time greatly when compared with UniXCoder.
>
> In the following Table, we observe that UniXcoder has not achieved better performance when compared to lexical retrievers. Besides, the results of Jaccard and BM 25 are comparable. Moreover, for Jaccard and BM25, we also benchmark the retrieval time consumption based on Intel Xeon CPU E5-2682 v4 with a base frequency of 2.50 GHz, where the results are 2.94ms and 3.14ms, respectively.  Therefore, we choose Jaccard as the default retriever in our M2rc-Eval.
>
> | **Model**          | **Jaccard** | **BM25**  | **Unixcoder** |
> | ------------------ | ----------- | --------- | ------------- |
> | StarCoder-7B       | 20.6/49.9   | 20.6/49.9 | 20.6/49.9     |
> | + Retrieval          | 23.6/49.3   | 23.4/49.8 | 22.9/49.7     |
> | + Retrieval & Tuning | 44.4/71.4   | 44.7/72.5 | 44.3/71.9     |

---

> ### Comment · Reviewer_PyQh · 2024-11-26
> **Thank You For The response**
>
> Thank you for getting back with the experiments regarding pertinent files in context. I would have been persuaded to raise my score had this been a method paper.
>
> However, the Java-only exec-based testing added during the review period in my view is not enough for a paper in the Datasets and Benchmarks track. This work would be ready for release if the authors were to take the effort to scale up synthetic test generation in all the languages and follow that up with human validation in a manner done by existing industry-standard benchmarks [1].
>
> I really believe that there is a niche for a massively multilingual repo-level code generation benchmark. Especially one that is execution tested. I hope the authors will rework and resubmit what I believe can be a very strong and well adopted benchmark but for now I have decided to retain the same score.
>
> ------------------------------------------------------------------------------------------------------------------------
>
> [1] Terry Yue Zhuo, et.al.: BigCodeBench: Benchmarking Code Generation with Diverse Function Calls and Complex Instructions. CoRR abs/2406.15877 (2024)

---

> > ### Author Response · Authors · 2024-11-30
> >
> > Dear Reviewer PyQh,
> >
> > Hello! Please refer to the updated results in  **General Response (G.Q2)**, where we have updated the execution analysis for 17 languages based on the follow-up suggestions.
> >
> > Besides, we believe that our submitted paper has improved a lot based on your insightful and constructive comments.
> >
> > Thanks again for your valuable efforts.

---

> ### Author Response · Authors · 2024-11-26
> **Thanks for your feedback.**
>
> Hi, thanks for your quick feedback. We have updated the results based on 17 languages in **General Response (G.Q2)**.  Note that HTML language cannot be executed, and we do not provide execution test samples for HTML.
>
> If any concerns remain, we would be grateful for further clarification and are happy to continue the discussion in this rebuttal process.

---

> ### Comment · Reviewer_PyQh · 2024-12-02
> **Thank You for the Response**
>
> This paper has improved during the review process. With the understanding that the authors will incorporate the 17 language collection, filtering and evaluation details into the final version, I have raised my score to 6.

---

### Official Review · Reviewer_TS5v · 2024-11-04

**Soundness:** 2
**Presentation:** 3
**Contribution:** 2
**Rating:** 3
**Confidence:** 5

**Summary:**

The paper introduces M$^2$RC-EVAL, a benchmark for repository-level code completion covering 18 programming languages, alongside M$^2$RC-INSTRUCT, a multilingual instruction dataset built in a similar way with M$^2$RC-EVAL. Using The Stack v2 dataset, the authors select repositories with 5+ stars containing 10-50 files, creating a dataset with 50,000 files per language for instruction tuning and 600 files per language for evaluation (w/ random 100 examples for validation and 500 for test). The benchmark introduces two types of annotations: bucket-level, an indicator of the depth of the AST node, and semantic-level, which was split in 11 major categories with language-specific subcategories. The authors benchmarked three major code LLMs (StarCoder-7B, DeepSeekCoder-6.7B, Code Llama-7B) under different settings and provide extensive analysis across multiple dimensions.

**Strengths:**

- This is the first dataset that covers 18 programming languages in repo-level code completion task
- The dataset Includes both evaluation and instruction tuning datasets
- Authors have conducted various evaluation across multiple dimensions

**Weaknesses:**

The most critical weakness of this paper is the use of The Stack v2 without addressing data contamination issues. Unlike previous work like Ding et al., 2023 which carefully controlled the date range to avoid overlap with model training data, this work directly uses a widely-used training corpus that contain data that was written many many years back. This fundamentally compromises the benchmark's ability to reflect real-world performance. The authors should either implement temporal splits or explicitly verify and control for overlap with model training data.

Further, the assumption that "we randomly choose a node on the AST as the completion cursor position" (Line 152) is overly restrictive and doesn't reflect real-world scenarios. Code completion can and does occur at any position in practice. A more realistic approach would be to allow arbitrary completion cursor positions while ensuring the completed code forms a valid AST node, similar to previous work cited in the paper.

In addition, the bucket-level annotation scheme appears arbitrary and lacks clear motivation. Dividing ASTs into exactly 10 buckets seems like an artificial constraint when simpler alternatives exist, such as directly using the node's layer number in the AST. It'd be great if authors could justify why this particular granularity was chosen or demonstrate its advantages over simpler approaches.

Finally, despite covering 18 programming languages, there was surprisingly very little discussion of language-specific insights. This is a great opportunity that the authors could seize to analyze how different language features, paradigms, and structures affect completion performance. Such analysis could provide valuable insights for both researchers and practitioners.

**Questions:**

In addition to the points mentioned in `Weaknesses`, I have a few questions and notes regarding the work:

1. Could the authors elaborate on how was the 5993 repos (line 204) split across eval and instruct?

2. It appears that adding retrieval seems to improve on EM but degrade on ES across models in Table 2. Why would this be expected?

3. On line 445: “StarCoder-7B base model already demonstrates strong coding proficiency, but lacks robust instruction-following capabilities” this is well expected as StarCoder-7B is a base model that has not been instruction tuned?

4. Some plots like Figure 4 look fancy but is conveying only little meaningful information in a readable way. Authors may consider using an alternative way to represent the data, e.g., a simple list or table. Similarly, Figure 5 is barely readable.

---

> ### Author Response · Authors · 2024-11-24
>
> **Q1: Data contamination issues.**
>
> **A1**: Thanks for your advice. Please See **General Response (G.Q1).**
>
>
>
> **Q2: A more realistic approach would be to allow arbitrary completion cursor positions while ensuring the completed code forms a valid AST node, similar to previous work cited in the paper.**
>
> **A2**: Thanks for your comments. As mentioned in many works [1, 2], developers often expect LLMs to complete the current code into a complete snippet, such as a completed code line or loop block, instead of suggesting an incomplete code snippet. Besides, the recent Qwen2.5-Coder adopts a similar way with M2rc-Instruct to produce the instruction dataset. Specifically, in Section 4.2 of Qwen2.5-Coder, the authors mentioned that ``Inspired by programming language syntax rules and user habits in practical scenarios, we leverage the tree-sitter-languages to parse the code snippets and extract the basic logic blocks as the middle code to infill''. Meanwhile, to demonstrate the effectiveness of our M2rc-Instruct, we also constructed a test dataset called M2rc-Eval (Random) based on your comments. Specifically,  for a fair comparison, based on the same repositories of M2rc-Eval, we randomly choose arbitrary completion cursor positions while ensuring the completed code forms a valid AST node, where the generated testing set is named M2rc-Eval (Random). The evaluation results  are as follows:
>
> | Model              | EM/ES (M2rc-Eval) | EM/ES (M2rc-Eval (Random)) |
> | ------------------ | ----------------- | -------------------------- |
> | StarCoder-7B       | 21.0/52.0         | 24.0/53.9                  |
> | + Retrieval          | 24.1/50.0         | 30.1/55.0                  |
> | + Retrieval & Tuning | 44.5/72.2         | 54.5/76.8                  |
> | DeepSeekCoder-6.7B | 22.6/54.7         | 25.3/56.2                  |
> | + Retrieval          | 25.1/51.7         | 32.4/59.6                  |
> | + Retrieval & Tuning | 46.8/74.1         | 55.7/78.3                  |
>
> We observe that higher performance is obtained in M2rc-Eval (Random). For this phenomenon, the possible reason is as follows. In each completion span,  the completion cursor position of our default M2rc-Eval is the start position of the syntax node block, which means that no context can be used inside the syntax node block. In contrast, the completion cursor position of M2rc-Eval (Random) is the arbitrary position of the syntax node block, which means that there exist additional informative contexts inside the current syntax node block for better completion. In other words, in M2rc-Eval (Random), the additional contexts are inside the syntax node block and before the completion cursor position, which can decrease the completion difficulty and improve the completion quality.  In our new version, we have included the above discussion.
>
> [1]. Qwen2.5-Coder Technical Report
>
> [2]. aiXcoder-7B: A Lightweight and Effective Large Language Model for Code Completion
>
>
>
> **Q3: Justify why this particular bucket granularity (i.e., 10).**
>
> **A3**: We have analyzed the minimum, maximum, and average depths of these 18 programming languages, and observed that depths of AST for different languages are different greatly, where the minimum depth is from 1 to 7, and the maximum depth is from 23 to 51. Therefore, we cannot use the node's layer number in the AST as the fine-grained annotation well. Besides, we observe that the average depth of different languages is from 9.7 to 20.2, and the overall average depth is  14.6. In our implementation, to explain clearly, we just chose 10 as the bucket number. Note that we will provide the tree depth number for each completion sample, and users can select the corresponding bucket number freely.
>
> | Depth   | Objective-C | C++  | C    | Haskell | Ruby | Kotlin | Rust | C#   | PHP  |
> | ------- | ----------- | ---- | ---- | ------- | ---- | ------ | ---- | ---- | ---- |
> | Minimum | 3           | 3    | 4    | 1       | 2    | 1      | 5    | 3    | 7    |
> | Maximum | 32          | 32   | 41   | 28      | 45   | 34     | 51   | 42   | 51   |
> | Average | 11.3        | 11.3 | 13.8 | 10.9    | 13.7 | 14.9   | 20.2 | 16.4 | 17.7 |
>
> |         | Go   | Java | HTML | R    | JavaScript | TypeScript | Scala | Lua  | Python |
> | ------- | ---- | ---- | ---- | ---- | ---------- | ---------- | ----- | ---- | ------ |
> | Minimum | 1    | 1    | 2    | 1    | 1          | 3          | 3     | 2    | 1      |
> | Maximum | 40   | 46   | 46   | 23   | 31         | 51         | 51    | 51   | 38     |
> | Average | 13.6 | 14.3 | 14.0 | 9.7  | 13.4       | 18.3       | 17.5  | 16.1 | 15.7   |

---

> ### Author Response · Authors · 2024-11-24
>
> **Q4: Discussion of language-specific insights.**
>
> **A4**: We have classified 18 programming languages in M2rc-Eval into 5 programming paradigms and 9 application scenarios as follows:
>
> | Paradigm Types             | Languages                                             |
> | ------------------------- | ------------------------------------------------------------ |
> | Procedural       | 'C'                                                          |
> | Object Oriented  | 'C#', 'Java', 'Kotlin', 'Objective-C'                        |
> |   Multiple Paradigms    | 'C++', 'Go', 'JavaScript', 'Lua', 'PHP', 'Python', 'R', 'Ruby', 'Rust', 'Scala', 'TypeScript' |
> |Functional         | 'Haskell'                                                    |
> | Markup Language | 'HTML'                                                       |
>
> | Application Scenarios | Languages                            |
> | --------------------- | ------------------------------------ |
> | Mobile                | 'Kotlin', 'Objective-C'              |
> | Cross Platform        | 'Java'                               |
> | Desktop Application   | 'C#'                                 |
> | Web Frontend          | 'JavaScript', 'TypeScript', 'HTML'   |
> | Web Backend           | 'Go', 'PHP', 'Ruby', 'Rust', 'Scala' |
> | Scientific Computing  | 'Python', 'R'                        |
> | System & Software     | 'C', 'C++'                           |
> | Education & Research  | 'Haskell'                            |
> | Automation Scripts    | 'Lua'                                |
>
> Based on the above programming classification structure, we also report the **EM** results  as follows, and have the following observations:
>
> (1). For different programming paradigms, we observe that the markup language paradigm has the lowest performance, and the functional paradigm has the best performance. Besides, after tuning, the performance of the markup language paradigm improves greatly. We assume that the syntax rules for markup language are easy, and these code LLMs can quickly obtain these rules after tuning.
>
> (2). For different application scenarios, the performance varies significantly. Specifically, the Web Frontend and Scientific Computing have relatively low performance, which needs to be improved for existing code LLMs.
>
> (3). We observe these LLMs share some common strengths and weaknesses and we will continue to investigate more language-specific insights to better improve the code completion abilities of existing code LLMs.
>
> | **Model**          | Procedural | Object Oriented | Multiple Paradigms | Functional | Markup Language |
> | ------------------ | ------------------- | ------------------------ | ------------------ | ------------------- | ------------------------ |
> | StarCoder-7B       | 19.2                | 21.3                     | 21.1               | 25.1                | 17.2                     |
> | + Retrieval          | 23.7                | 24.2                     | 24.3               | 24.8                | 21.3                     |
> | + Retrieval & Tuning | 47.6                | 47.4                     | 43.4               | 44.6                | 46.7                     |
> | DeepSeekCoder-6.7B | 22.6                | 22.4                     | 23.0               | 24.4                | 18.6                     |
> | + Retrieval          | 31.0                | 30.1                     | 28.8               | 26.5                | 24.3                     |
> | + Retrieval & Tuning | 47.3                | 49.4                     | 46.1               | 44.6                | 46.3                     |
>
> | **Model**            | StarCoder-7B | +Retrieval |+Retrieval & Tuning|DeepSeekCoder-6.7B | +Retrieval | +Retrieval & Tuning |
> | -------------------- | ------------ | --------- | ------------------ | ------------------ | --------- | ------------------ |
> | Mobile               | 21.2         | 22.9      | 48.0               | 23.4               | 30.3      | 49.7               |
> | Cross Platform       | 24.2         | 25.3      | 48.1               | 22.4               | 38.1      | 49.2               |
> | Desktop Application  | 18.7         | 26.1      | 45.3               | 20.3               | 31.8      | 48.6               |
> | Web Frontend | 17.9         | 22.2      | 41.5               | 19.4               | 26.0      | 43.7               |
> | Web Backend        | 22.8         | 24.8      | 44.6               | 25.6               | 29.9      | 47.3               |
> | Scientific Computing | 18.2         | 23.5      | 40.3               | 20.6               | 28.0      | 44.2               |
> | System & Software    | 21.3         | 24.0      | 50.1               | 23.1               | 31.0      | 50.6               |
> | Education & Research | 25.1         | 24.8      | 44.6               | 24.4               | 26.5      | 44.6               |
> | Automation Scripts   | 21.7         | 26.3      | 43.7               | 20.2               | 26.7      | 43.5               |

---

> ### Author Response · Authors · 2024-11-24
>
> **Q5: Elaborate on repo splits (line 204) across eval and instruct.**
>
> **A5**: In Line 204, we only provide the number of repositories for testing split,  and the numbers of repositories for train, validation, and test sets are 37439, 1635, and 5993, respectively. We have updated this detail in our new version.
>
>
>
> **Q6: Degradation issue on ES for  StarCoder-7B, Code LLama-7B, DeepSeekCoder-6.7B.**
>
> **A6**: As shown in the **General Response (G.Q1)**, we observe that the recent strong general LLMs (LLama3.1-70B, Qwen2.5-72B, GPT-4o, Claude 3.5 Sonnet, DeepSeek-V2.5) and code LLMs (DeepSeekCoder-33B, Qwen2.5-Coder-7B, Qwen2.5-Coder-32B) usually obtain stable and sufficient improvements on both EM and ES metrics when introducing retrieval strategy. In contrast, the capabilities of original base models (e.g., StarCoder-7B, Code LLama-7B, DeepSeekCoder-6.7B) are relatively weaker, and the improvements are not stable. Notably, we observe that StarCoder-7B and   DeepSeekCoder-6.7B achieve better performance on both EM and ES metrics in M2rc-Eval-2403 and M2rc-Eval-2406 splits.
>
>
>
> **Q7: Issue of Line 445.**
>
> **A7**: Thanks for your pointing out this misleading issue. We have removed the corresponding description in our new version.
>
>
>
>
>
> **Q8: Better illustration of Figures 4 and 5.**
>
> **A8**: Thanks for your suggestions. For Figure 4  in the original version, we have replaced the figure with a table (Table 2) for better illustration. For Figure 5 in the original version, we have reduced the number of languages in Figure 4 and added a table (Table 6) for better illustration.

---

> ### Author Response · Authors · 2024-11-29
>
> Dear Reviewer TS5v,
>
> Hello! We believe we have addressed your concerns carefully. If you have other questions or comments, please let us know. We are very glad to solve your concerns.
>
> Thanks for your insightful suggestions.

---

> > ### Comment · Reviewer_TS5v · 2024-12-01
> >
> > Thanks for the detailed response. Most of my questions are addressed and please also see my ask under G.Q2.

---

> > > ### Author Response · Authors · 2024-12-02
> > >
> > > Dear Reviewer TS5v,
> > >
> > > As the discussion deadline is coming, please let us know whether our responses have addressed all your concerns. Besides, if you recognize that we have solved your questions well, could you reevaluate our work and change your rating?
> > >
> > >
> > > Moreover, we believe that our submitted paper has improved a lot based on your insightful and constructive comments.
> > >
> > > Thanks again for your valuable efforts.

---

> ### Author Response · Authors · 2024-12-02
>
> Thanks for your feedback.
> ﻿
> For the new questions, the responses are as follows:
> ﻿
>
> (1). As mentioned in the previous response, generating unit test cases and providing execution sandboxes for repository-level code completion are very challenging. Moreover, existing open-sourced sandboxes usually focus on file-level or function-level code execution and there are no execution sandboxes, which support repository-level execution. In our implementation of these 17 languages, we first prepare corresponding docker execution environments for the repositories for these 815 samples in G.Q2. Then, as the rebuttal phase is short, the authors of M2rc-Eval **manually** execute unit-test cases for these samples.
> ﻿
>
> (2). For the scalable issue of this benchmark,  as the benchmark usually includes a relatively small number of samples, we have prepared the docker files or requirements for each repository, which are essential for execution. In our future work, we will continue to investigate how to execute repository-level unit-test cases **automatically**. Notably, if we can automatically build the sandbox environment based on the provided docker files or requirements, we can easily scale up the benchmark size.
> ﻿
>
> (3). For the coverage issue, we think that it is not necessary to compute the coverage metric for repository-level code completion. Specifically, the coverage metric is usually used to measure the proportion of executed lines of code for each function body. However, the completion code is usually a part of the function body, where the number of code completion lines is usually less than 5. In this way, in the process of collecting test cases in G.Q2, we ensure that **the test cases execute all code completion lines**, and we do not need to keep all lines in each function body executed.
>
> We will add the above details in our new version.
>
> If you still have questions, please let us know. We are glad to solve your concerns carefully.

---

> ### Author Response · Authors · 2024-12-03
>
> Dear Reviewer TS5v,
>
> As the discussion deadline will end soon, please let us know whether our responses have addressed all your new questions. Moreover, as you have mentioned that we have addressed most of your questions, could you reevaluate our work and improve your rating?
>
> Additionally, we believe that our M2rc-Eval has improved a lot based on your insightful and constructive suggestions.
>
> Thanks again for your valuable efforts.

---

> > ### Author Response · Authors · 2024-12-03
> >
> > Hi, Reviewer TS5v,
> >
> > Thank you for your valuable comments. As the discussion period is coming to a close, we would appreciate it if you could let us know whether our responses have addressed your concerns.

---

### Official Review · Reviewer_uHra · 2024-11-04

**Soundness:** 4
**Presentation:** 4
**Contribution:** 2
**Rating:** 5
**Confidence:** 4

**Summary:**

The paper introduces M2rc-Eval, a new benchmarks comprising repository-level code completion tasks in many more languages than current benchmarks. Unique features of this benchmark include fine-grained annotations for different types of completion scenarios. The authors also introduce M2rc-Instruct, which a training dataset to improve performance on M2rc-Eval.

**Strengths:**

- **Useful contributions.** The idea of having a very-large scale benchmark for repository-aware code completions across 18 programming languages is very useful. This will significantly help researchers improving Code LLMs on low resource languages such as Lua and Scala.
- **Novelty.** The fine-grained breakdown of code completion tasks is novel. However, I am not sure if this is needed or insightful. I would encourage the authors to present more analyses around this.
- **Interesting analyses.** I found the multi-lingual transfer experiment to be quite interesting and would be interested in a language-wise breakdown of this.
- **Paper exposition.** The paper is well-written and the analyses are presented in an easy-to-understand manner.

**Weaknesses:**

- **Data leakage concerns.** The presented benchmark is collected from The Stack v2 which sources code from public sources such as GitHub. As such it may have significant overlap with the training corpora of the Code LLMs already evaluated in the paper and for future LLMs. As such, this benchmark is not fool-proof from future leakage and the utility of the benchmark for future evaluations is questionable. Further, the authors have evaluated only small models, and evaluations on bigger models ≥ 15B or strong models like GPT-4o or Claude 3.5 Sonnet might reveal the effect of leakage more concretely.
- **Repository-level coding.** While the paper says that the benchmark is for repository-level code completion evaluations, the data collection described in Section 3.1 does not guarantee that the tasks in the benchmark strictly require repository-level contexts.
- **Evaluation metrics.** While EM/ES metrics are being used for evaluations and have been used in several previous works as well, I think it is time that we move on from these inaccurate metrics. We should resort to more robust metrics relying on static or execution analysis such as pass rate [1] or hallucination rate [2] for measuring correctness of code completions.
- **Nitpicks.**
    - The analyses in lines 416-428 are quite obvious, i.e., there is a negative correlation between ES and the number of tokens generated
    - Typos:
        - 047 - “can not” → cannot
        - 261 - textural → textual
        - 401 - shadow → shallow

[1] Chen, Mark, et al. "Evaluating large language models trained on code." *arXiv preprint arXiv:2107.03374* (2021).

[2] Jain, Nihal, et al. "On Mitigating Code LLM Hallucinations with API Documentation." *arXiv preprint arXiv:2407.09726* (2024).

**Questions:**

See comments above. But framed some concerns in question format below:

1. How do larger or more recent models such as StarCoder2-15B or GPT-4o perform on this benchmark?
2. Can you provide more insights from your multi-lingual transfer experiments? Specifically which languages benefit the most from Python-only fine-tuning? Can we predict such transfer ahead of time?

---

> ### Author Response · Authors · 2024-11-24
>
> **Q1: Data leakage concerns.**
>
> **A1**: Please See **General Response (G.Q1)**.
>
>
>
> **Q2: Repository-level coding.**
>
> **A2**: In Lines 209-210 of Section 3.2, we discard test samples that could be exactly predicted by DeepSeekCoder-1.3B without cross-file contexts. Meanwhile, to discuss more clearly, we also use three strong code LLMs (i.e., DeepSeekCoder-6.7B, StarCoder-7B, and DeepSeekCoder-33B) to analyze the ratios of evaluation cases with or without using repository-level contexts. Specifically, we prompt DeepSeekCoder-6.7B, StarCoder-7B, and DeepSeekCoder-33B using the in file contexts of each sample and obtain three predictions. Then, if one prediction exactly matches the ground truth, this sample is considered to be predicted without requiring repository-level contexts. Finally, we observe that 71% of samples cannot be well predicted only using in file contexts, which indicates that our dataset is challenging. We have added this detail in our new version in the Appendix A.6.
>
>
>
> **Q3: Evaluation metrics.**
>
> **A3**: Thanks for your suggestions. Please See **General Response (G.Q2)**.
>
> We have cited these provided works [1] [2] and updated the above discussion in our new version.
>
> [1] Chen, Mark, et al. "Evaluating large language models trained on code." arXiv preprint arXiv:2107.03374
>
> [2] Jain, Nihal, et al. "On Mitigating Code LLM Hallucinations with API Documentation."arXiv:2407.09726
>
>
>
> **Q4: Analyses in lines 416-428 are quite obvious.**
>
> **A4**: Thanks for your suggestions. We have rewritten the analyses by removing useless analyses in our new version.
>
>
>
> **Q5: Typos.**
>
> **A5**: We have revised these typos in our new version.
>
>
> **Q6: Evaluation of large models.**
>
> **A6**: Please See **General Response (G.Q1)**.

---

> ### Author Response · Authors · 2024-11-24
>
> **Q7: More insights from multi-lingual transfer experiments.**
>
> **A7**: We have provided the detailed results of different languages in the new version (Fig. 22 and Fig. 23). Below, we also provide the results of relative improvements after using Python-only tuning, and the observations are as follows. (1) We observe that the improvement of HTML is the highest.  For this phenomenon, we assume that HTML is a markup language with a few simple syntax rules, which is easy to learn. (2). The improvement in Python language is significant using Python-only tuning, which is straightforward. (3). Most high-resource languages (e.g., Go, C++, C#, Scala) benefit a lot based on python-only tuning. For this phenomenon, as code LLMs have good fundamental capacities for these high-resource languages due to the large pretraining quota, it may be easy to promote the emergent code completion abilities of other languages based on the strong fundamental abilities of these high-resource languages. (4). The improvements on these low-resource languages (e.g., Rust, Haskell, Lua) are usually relatively less than the improvements on high-resource languages.
>
> | Language    | + Retrieval | + Retrieval & Tuning (Python Only) | Relative Improvements |
> | ----------- | ----------- | ---------------------------------- | --------------------- |
> | HTML        | 44.01%      | 67.10%                             | 52.47%                |
> | Python      | 45.43%      | 65.50%                             | 44.18%                |
> | Go          | 51.70%      | 74.15%                             | 43.42%                |
> | C++         | 47.89%      | 68.62%                             | 43.29%                |
> | C#          | 50.24%      | 71.39%                             | 42.10%                |
> | Scala       | 45.02%      | 63.81%                             | 41.74%                |
> | JavaScript  | 51.96%      | 72.06%                             | 38.68%                |
> | C           | 50.64%      | 69.89%                             | 38.01%                |
> | R           | 52.87%      | 72.79%                             | 37.68%                |
> | Objective-C | 44.43%      | 60.89%                             | 37.05%                |
> | Java        | 51.93%      | 70.78%                             | 36.30%                |
> | Kotlin      | 49.92%      | 67.84%                             | 35.90%                |
> | TypeScript  | 51.33%      | 69.72%                             | 35.83%                |
> | PHP         | 52.62%      | 71.11%                             | 35.14%                |
> | Lua         | 46.06%      | 60.86%                             | 32.13%                |
> | Ruby        | 45.13%      | 59.30%                             | 31.40%                |
> | Rust        | 56.14%      | 72.90%                             | 29.85%                |
> | Haskell     | 49.61%      | 62.57%                             | 26.12%                |
> | Overall     | 49.27%      | 67.85%                             | 37.70%                |
>
> Finally, it should be mentioned that the cross-lingual transfer abilities are influenced by many factors (e.g., fundamental abilities of base LLMs for different languages, syntax similarities among different languages, data qualities, and quantities of tuning stage). Therefore, we think that it is not easy to predict well ahead of time. In the future, we will continue to investigate the cross-lingual scaling law among different languages for both pretraining and tuning stages to better explain the cross-lingual transfer abilities for code intelligence.

---

> > ### Comment · Reviewer_uHra · 2024-12-02
> > **Reviewer Response**
> >
> > Thanks for your detailed comments and additional experiments!
> >
> > I would like to retain my original score. Based on my understanding, the paper's main net-new contribution is a multi-lingual benchmark for repository-aware coding. The experiments, especially on multi-lingual transfer of capabilities, are relevant but do not provide sufficiently new insights. There are many papers touching upon repository-aware coding and fine-tuning models for this task. The new insights that the paper introduce are unclear to me. I believe the paper would benefit from stating their goals as creating the dataset and clearly outlining what new insights the evaluations on this dataset brought, other than being a new benchmark for repository-aware coding.
> >
> > Additional nit: The table in the general response shows GPT-4o and Claude 3.5 Sonnet to be worse than much smaller models like StarCoder-7B. This is extremely unlikely and I recommend the authors to re-check their evaluations.

---

> > > ### Author Response · Authors · 2024-12-02
> > >
> > > Thanks for your feedback.
> > >
> > > For the results of GPT-4o and Claude 3.5 Sonnet, we need to clarify the following details:
> > >
> > > (1). The open-sourced code-specific LLMs (e.g., StarCoder-7B) **use the FIM (Fill-in-the-Middle) training loss in the pre-training process**, which uses the prefix and suffix code contexts to predict the middle part. Correspondingly, for repository-level code completion, we also use the prefix and suffix code contexts to complete the middle code block. Therefore, the existing code LLMs can achieve relatively better performance with the help of FIM training.
> > >
> > > (2). For closed-sourced general LLMs (e.g., GPT-4o and Claude 3.5 Sonnet), these LLMs are usually produced for general understanding, and the code-related abilities are usually enhanced for code question-answer tasks, where **the FIM pattern is not well supported in these LLMs**. Therefore, we need to use the prompt engineering strategy to obtain the repository-level code completion results, where the prompt template is provided G.Q1. Moreover, we have run the results on close-sourced general LLMs three times and observed the results are relatively stable.
> > >
> > > (3). Previous work CrossCodeEval [1] reports the results of **GPT 3.5** on the repository-level code completion dataset (CrossCodeEval), where the results **are also weaker a lot when compared with the code-specific LLMs** (e.g., CodeGen25-7B, StarCoder-15.5B).
> > >
> > >
> > > Therefore, we believe that **the results on GPT-4o and Claude 3.5 Sonnet are correct**.
> > >
> > > [1]. CrossCodeEval: A Diverse and Multilingual Benchmark for Cross-File Code Completion, NeurIPS 2023 Datasets and Benchmarks
> > >
> > >
> > > For the multilingual insights, we have provided other results on language-specific insights (See Reviewer TS5v.Q4), and we will continue to investigate more insights in our new version.
> > >
> > >
> > > Finally, we believe that our submitted paper has improved a lot based on your insightful and constructive comments.
> > >
> > > Thanks again for your valuable efforts.

---

> ### Author Response · Authors · 2024-11-30
>
> Hi, Reviewer uHra,
>
> We believe we have addressed your concerns carefully. If you have other questions or comments, please let us know. We are very glad to solve your concerns.
>
> Thanks for your insightful suggestions.

---

> > ### Author Response · Authors · 2024-12-02
> >
> > Hello, Reviewer **uHra**,
> >
> > Thanks again for your insightful comments. As the discussion deadline is coming, please let us know if our responses have addressed your concerns well.

---

### Author Response · Authors · 2024-11-24
**General Response**

# General Response
#### **G.Q1: Data leakage concerns and evaluation on more strong models.**

**A1**: Following LiveCodeBench and EvoCodeBench, after submission, we also build a dynamically updating M2rc-Eval dataset, where the M2rc-Eval-2403 and M2rc-Eval-2406 are produced.

Specifically, we collect repositories from 2024.03.01-2024.05.31 and then build the **M2rc-Eval-2403** split based on the data collection process of Sections 3.1 and 3.2. Similarly, we build the **M2rc-Eval-2406** using repositories from 2024.06.01-2024.08.30. In the following table, we report the results (EM/ES) of different splits for different LLMs (Code Llama-7B, StarCoder-7B, DeepSeekCoder-6.7B, GPT-4o, LLama3.1, Qwen2.5, Claude 3.5 and DeepSeek-V2.5). Note that as some models (i.e., general LLMs) do not support the FIM pattern, we directly use prompt engineering strategy to obtain the repository-level code completion results, where the prompt template is provided in our new version.

We have the following observations. (1). When introducing cross-file context using retrieval, better performance results are usually obtained for both code LLMs and General LLMs. (2). For existing LLMs, the performance on different testing splits (M2rc-Eval, M2rc-Eval-2403, M2rc-Eval-2406) are relatively stable, which means that data leakage or contamination concern does not have a significant impact on evaluating the code completion abilities in M2rc-Eval. Besides, for many knowledge-based benchmarks (e.g., MMLU, SimpleQA),  this knowledge information widely exists in web and book corpus, which have been trained in existing LLMs. However, these benchmarks are still effective tools for evaluating the knowledge coverage degree in existing LLMs. (3) Meanwhile, although our M2rc-Eval has been trained in several LLMs, we still find existing LLMs cannot achieve competitive performance results, and our M2rc-Eval can still be used as an effective benchmark to evaluate the code completion abilities of existing LLMs. (4) These powerful API LLMs or open-source LLMs (e.g., LLama3.1-70B, Qwen2.5-72B) have strong code generation abilities in many benchmarks (e.g., HumanEval, MBPP, LiveCodeBench), but the repository-level code completion results are still limited when compared to these code-specific LLMs. We assume that these code LLMs usually introduce an FIM loss objective in training, which is the same as the testing scenes and greatly improves the repository-level code completion.

| **Model**          | **M2rc-Eval** | **M2rc-Eval-2403** | **M2rc-Eval-2406** |
| ------------------ | ------------- | ------------------ | ------------------ |
| **Code LLMs**      |               |                    |                    |
| Code Llama-7B      | 19.4/50.3     | 19.1/52.9          | 21.5/52.7          |
|  + Retrieval          | 20.2/46.1     | 23.1/50.8          | 25.0/51.5          |
| StarCoder-7B       | 21.0/52.0     | 20.4/53.1          | 20.1/51.6          |
|  + Retrieval          | 24.1/50.0     | 26.0/54.9          | 28.6/55.9          |
| StarCoder2-15B       | 23.2/53.6     | 22.4/52.7          | 22.9/52.7          |
|  + Retrieval          | 25.3/54.2     | 26.2/55.1          | 25.9/54.8          |
| DeepSeekCoder-6.7B | 22.6/54.7     | 20.4/51.9          | 23.0/55.6          |
|  + Retrieval          | 25.1/51.7     | 24.0/52.7          | 30.3/56.4          |
| DeepSeekCoder-33B  | 26.8/51.6     | 24.0/43.7          | 23.9/49.7          |
| + Retrieval          | 27.3/52.9     | 27.1/51.8          | 27.5/49.8          |
| Qwen2.5-Coder-7B         | 18.8/46.5     | 20.5/49.7          | 21.0/48.1          |
| + Retrieval          | 27.2/52.2     | 31.0/57.2          | 32.4/56.7          |
| Qwen2.5-Coder-32B        | 34.7/65.7     | 35.0/66.2          | 37.3/67.6          |
| + Retrieval          | 41.7/68.0     | 43.9/69.5          | 45.9/71.2          |
| **General LLMs**   |               |                    |                    |
| LLama3.1-70B       | 6.4/31.9      | 5.0/31.5           | 5.4/31.3           |
| + Retrieval          | 6.8/33.0      | 6.1/33.3           | 6.1/32.8           |
| Qwen2.5-72B        | 6.7/39.1      | 11.6/49.6          | 10.2/45.3          |
| + Retrieval          | 12.2/44.8     | 12.4/51.1          | 13.4/50.9          |
| GPT-4o             | 12.2/45.5     | 11.5/54.0          | 11.1/47.2          |
| + Retrieval          | 17.8/56.7     | 15.0/57.4          | 17.3/54.0          |
| Claude 3.5 Sonnet  | 22.4/55.3     | 23.2/63.8          | 23.1/59.5          |
| + Retrieval          | 29.9/62.8     | 28.4/65.9          | 30.5/67.1          |
| DeepSeekV2.5       | 16.1/50.5     | 23.9/61.0          | 25.2/56.9          |
| + Retrieval          | 27.2/60.6     | 28.3/64.1          | 26.0/61.1          |

We have added this discussion in our new version.

---

### Author Response · Authors · 2024-11-24
**General Response**

# General Response



#### **G.Q2: More evaluation metrics.**

**A2**: Thanks for your suggestions. We have additionally provided the syntax static analysis and execution analysis.

For syntax static analysis, following Qwen2.5-Coder, to further verify the syntax correctness of the predicted code snippets, we use the code static checking tools (Tree-Sitter) for all predicted code snippets of test split of M2rc-Eval. Specifically, we parse the code snippet into the abstract syntax tree and filter out the code snippet, where the parsed nodes in the code snippet have parsing errors.

The results for syntax static analysis are as follows, and we observe that syntax accuracy improves a lot after tuning. Notably, the syntax accuracy is close to 100% after tuning, which means that existing code LLMs can easily learn the basic syntax rules for existing programming languages.

| Metrics                       | Code Llama-7B (+ Retrieval) | Code Llama-7B( + Retrieval & Tuning) | StarCoder-7B(+ Retrieval) | StarCoder-7B( + Retrieval & Tuning) | DeepSeekCoder-6.7B(+ Retrieval) | DeepSeekCoder-6.7B( + Retrieval & Tuning) |
| ----------------------------- | --------------------------- | ------------------------------------ | ------------------------- | ----------------------------------- | ------------------------------- | ----------------------------------------- |
| Syntax static analysis (Acc.) | 71.3                        | 96.1                                 | 83.9                      | 96.9                                | 80.4                            | 97.2                                      |



For execution analysis, as discussed in Appendix A.4, generating unit test cases and providing execution sandboxes for repository-level code completion are very challenging. In this rebuttal phase, we follow RepoCoder to provide the execution test samples in 17 languages. Note that HTML language cannot be executed, and we do not provide execution test samples for HTML. Specifically, as running tests can be time-consuming and computationally expensive, we first randomly select a separate set of smaller-scale repositories that are easy to deploy. Besides, as collecting unit tests is challenging, we directly utilize the unit tests available in these repositories and annotate corresponding functions covered by these unit tests. Finally, we utilize unit tests present in the repository to evaluate the functional correctness of the completed function body, where we report the Pass@1(Pass rate is 1 if the code passes all the corresponding test cases, and 0 otherwise). Note that the number of samples for execution analysis is 815.  The average execution accuracy results in 17 languages are as follows, and we observe that retrieval and tuning on our M2rc-Instruct both lead to better performance.

| Model                | Avg Acc. |
| -------------------- | ---- |
| StarCoder-7B         | 29.9 |
| + Retrieval          | 32.3 |
| + Retrieval & Tuning | 56.6 |
| DeepSeekCoder-6.7B   | 33.7 |
| + Retrieval          | 39.1 |
| + Retrieval & Tuning | 60.5 |

---

> ### Comment · Reviewer_TS5v · 2024-12-01
>
> Thanks the authors for the detailed response.
>
> Could you please clarify how you were able to do execution in 17 languages? Different languages require different ways to build and test, how would you make it scalable enough as a benchmark? Further, how would you validate that these unit tests have high coverage that could ensure the reliability of the result?

---

> > ### Author Response · Authors · 2024-12-02
> >
> > Thanks for your feedback.
> >
> > Please see the new response in the review section of **Reviewer TS5v**.

---

### Author Response · Authors · 2024-11-24
**General Response**

**Prompt template for G.Q1.**

The prompt template for **G.Q1** for evaluation on some general LLMs, which do not support the FIM pattern.


```
## First, input a segment of <LANG> code that needs completion. Please help complete the code at the corresponding position.

## The format of the input code is as follows:
<fim_start> prefix <fim_hole> suffix <fim_end>

Explanation:
1. <fim_start>, <fim_hole>, and <fim_end> are special characters.
2. <fim_hole> is the position that needs completion.
3. The prefix after <fim_start> represents the context before the content that needs completion.
4. The suffix after <fim_hole> represents the context following the content that needs completion.

## The output format is as follows:
    1. Only the code completion result for the position <fim_hole> is needed.
    2. Do not use markdown format.
    3. Do not include the surrounding context.
    4. Do not provide any explanation or description.

## The content of the input code is as follows:
<CODE>

---

### Author Response · Authors · 2024-11-25
**Summarization on the Responses**

Thanks for handling/reviewing our submitted manuscript: "M2rc-Eval: Massively Multilingual Repository-level Code Completion Evaluation". We would like to thank the reviewers for their insightful and constructive comments and suggestions. By addressing each of the issues raised by the reviewers, we believe that the quality and clarity of our M2rc-Eval can be improved a lot. The major responses are summarized as follows:

(1). We have carefully discussed the data leakage concerns by introducing two newly updated test sets based on our data collection process (See  Reviewer uHra.Q1, Reviewer TS5v.Q1).

(2). We have additionally reported the results of recent code-specific LLMs and the powerful general LLMs (See Reviewer uHra.Q6, Reviewer PyQh.Q3, Reviewer 5s7R.Q3).

(3). We have provided more evaluation results based on syntax static analysis and execution analysis (See Reviewer uHra.Q3, Reviewer PyQh.Q2, Reviewer 5s7R.Q1).

(4). We have provided more clarification and discussion on data collection process and multi-lingual transfer (See Reviewer uHra.Q2&Q4&Q7).

(5). We have discussed more details on another data construction strategy, justification on the bucket granularity, language-specific insights, and degradation for several LLMs (See Reviewer TS5v.Q2&Q3&Q4&Q6).

(6). We have discussed more details on evaluation setup, results based on long-context LLMs, and the effect of different retrievers (See Reviewer PyQh.Q1&Q4&Q6).

(7). We have discussed the correlation between CodeBLEU and EM/ES, and qualitative error analysis (See Reviewer 5s7R.Q2&Q4).

(8). We have significantly improved the writing quality based on comments of typos and figure presentation (See Reviewer uHra.Q5, Reviewer TS5v.Q5&Q7&Q8, Reviewer PyQh.Q5, Reviewer 5s7R.Q5).

Again, we would like to sincerely thank you very much for these constructive comments and evaluation for our manuscript.

---

### Author Response · Authors · 2024-11-25
**Looking forward to feedback on the Responses.**

Dear Reviewers:

Hello! We have updated the responses to your constructive and insightful comments, and we would like to kindly ask you to take a look at our responses and reevaluate our work based on our clarifications. Please let us know whether our response addresses your concerns or whether there is any further detail we can provide to help address them. We appreciate your time and consideration!

---

### Author Response · Authors · 2024-11-30

Dear Reviewers,

Hello! Since the discussion period is short, we kindly ask you to review our responses and reevaluate our work given our clarifications.  If any concerns remain, we would be grateful for further clarification and are happy to continue the discussion in this rebuttal process.

We appreciate your valuable time and consideration!

Thanks.

---

### Meta-Review · Area_Chair_S9Zc · 2024-12-16

**Metareview:**

The paper creates a repo-level code completion benchmark across 18 programming languages. The benchmark introduces two types of annotations: bucket-level, an indicator of the depth of the AST node, and semantic-level, which was split in 11 major categories with language-specific subcategories. The evaluation benchmarks three major open-weight code LLMs (StarCoder-7B, DeepSeekCoder-6.7B, Code Llama-7B) under different settings and provide extensive analysis across multiple dimensions.

While the data resource is appreciated, the main concern lies in the design of evaluation metric. The work mostly focuses surface form matching, which has shown to not have strong correlation with functional correctness in previous works. Although the authors provide results on execution-based evaluation for a small subset of the data during rebuttal, it does not cover the full evaluation set and hence cannot be considered as part of the work

**Additional Comments On Reviewer Discussion:**

NA

---

### Decision · Program_Chairs · 2025-01-22

Reject